# Tumor phylogeography reveals block-shaped spatial heterogeneity and the mode of evolution in Hepatocellular Carcinoma

Xiaodong Liu [1,2,12], Ke Zhang [3,12], Neslihan A. Kaya [4], Zhe Jia[3], Dafei Wu[1], Tingting Chen[1,5], Zhiyuan Liu [1,5], Sinan Zhu[1,6], Axel M. Hillmer [7], Torsten Wuestefeld [4,8], Jin Liu [6], Yun Shen Chan[9], Zheng Hu [10], Liang Ma [1,13] ✉, Li Jiang[3,13] ✉ & Weiwei Zhai [1,11,13] ✉

Solid tumors are complex ecosystems with heterogeneous 3D structures, but the spatial intra-tumor heterogeneity (sITH) at the macroscopic (i.e., whole tumor) level is under-explored. Using a phylogeographic approach, we sequence genomes and transcriptomes from 235 spatially informed sectors across 13 hepatocellular carcinomas (HCC), generating one of the largest datasets for studying sITH. We find that tumor heterogeneity in HCC segregates into spatially variegated blocks with large genotypic and phenotypic differences. By dissecting the transcriptomic heterogeneity, we discover that 30% of patients had a "spatially competing distribution" (SCD), where different spatial blocks have distinct transcriptomic subtypes co-existing within a tumor, capturing the critical transition period in disease progression. Interestingly, the tumor regions with more advanced transcriptomic subtypes (e.g., higher cell cycle) often take clonal dominance with a wider geographic range, rejecting neutral evolution for SCD patients. Extending the statistical tests for detecting natural selection to many non-SCD patients reveal varying levels of selective signal across different tumors, implying that many evolutionary forces including natural selection and geographic isolation can influence the overall pattern of sITH. Taken together, tumor phylogeography unravels a dynamic landscape of sITH, pinpointing important evolutionary and clinical consequences of spatial heterogeneity in cancer.

Tumors are complex ecosystems organized into heterogeneous three-dimensional structures, how tumor and non-tumor cells distribute and evolve spatially is a central question in tumor evolution and treatment response[1,2]. Even though intratumor heterogeneity (ITH) has been extensively characterized across many cancer types[3,4], previous studies often surveyed generic ITH from the limited number of sectors (e.g., $n = 3$–$5$) without in-depth exploration of spatial intra-tumor heterogeneity (sITH)[5,6]. Although spatial sampling has been employed to address specific biological questions such as phenotypic differences between central and peripheral regions[7], microenvironmental spatial

heterogeneity[8–10] as well as the evolutionary origin of metastasis[7,11], the overall landscape of sITH across cancer types are largely unknown. In addition, previous work often surveyed the ITH at the genetic level, understanding the joint evolution of genotypic and phenotypic ITH at the macroscopic (i.e., whole tumor) level remains understudied.

Parallel to genomic surveys of ITH, many evolutionary models including linear[12], branched[1,4,13], neutral[14,15], and punctuated evolution[16–19] have been proposed for tumor evolution. However, none of these models are explicit regarding the spatial organization of ITH. Even though computational modeling of tumors as spatially expanding

populations has been extensively explored[6,7,14,20–25], the gap between theoretical modeling and empirical data is still quite large[26]. As shown in recent computational simulations[20,22], the mode of tumor evolution can be heavily influenced by the spatial organization of tumor cells (i.e., tissue architecture)[20–22,27] varying across different tissue types[24]. Characterizing sITH and studying the mode of spatial tumor evolution is pivotal for understanding the history of tumor evolution.

In this work, we employ a phylogeographic approach and sample exomes and transcriptomes extensively from 235 sectors across 13 HCC patients, generating one of the largest datasets for studying sITH. By integrating physical coordinates of tumor sectors with genomic and transcriptomic information, we dissect spatial distributions of distinct subclones and depict their evolution at the genotypic and phenotypic levels. We find that tumor heterogeneity in HCC segregates into spatially variegated blocks with large genotypic and phenotypic differences. By dissecting the transcriptomic heterogeneity, we discover that 30% of patients had a "spatially competing distribution" (SCD), where different spatial blocks have distinct transcriptomic subtypes co-existing within a tumor, capturing the critical transition period in disease progression. Population Genetic analysis unravel a strong signal of natural selection in SCD

patients suggesting the importance of adaptive evolution in driving disease progression. Taken together, we unravel a dynamic landscape of sITH and pinpoint important evolutionary and clinical consequences of spatial heterogeneity.

## Results

### A phylogeographic approach for surveying sITH

Thirteen patients who underwent surgical treatment for HCC were recruited for this study. As a surgical cohort, most of these patients have early-stage disease and are all hepatitis B positive (Supplementary Data 1). Even though several multi-regional approaches have been conducted for HCC (e.g., recently reviewed in Craig et al.[28]), the number of sectors taken was often 3–5 and the sITH has not been sufficiently explored. In order to systematically survey the sITH across the whole tumor, we sampled multiple sectors from a central slice of the tumor in a honeycomb manner, resulting in 5–30 sectors (mean = 17) depending on the size of the tumor (Fig. 1a and Supplementary Fig. 1). Adjacent normal liver tissues were also harvested as the genomic control. In total, we have taken 222 tumor sectors and 13 normal samples, resulting in one of the largest datasets for studying sITH in HCC and across cancer types[7,21].

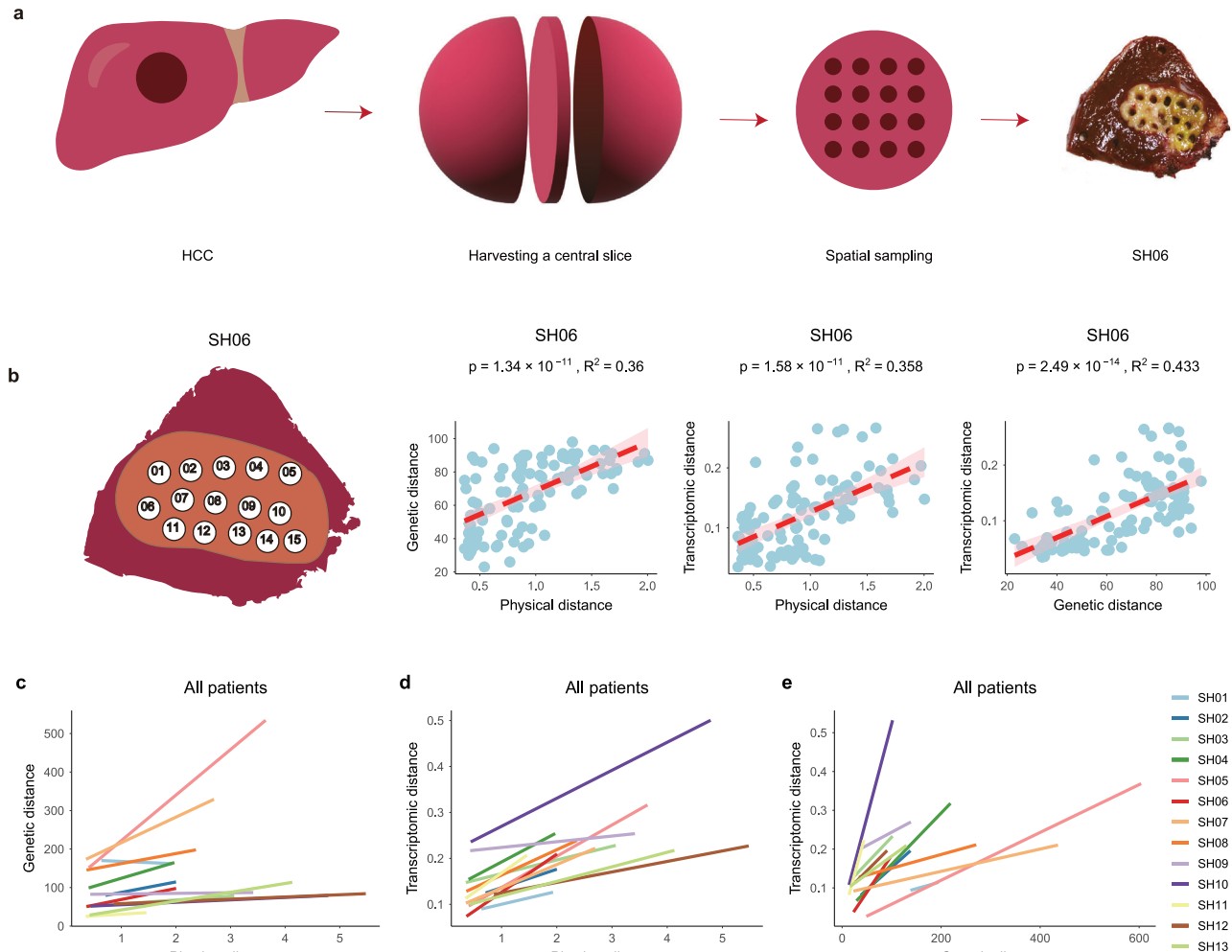

Fig. 1 | The cellular phylogeographic sampling and the IBD pattern. a The workflow of the phylogeographical sampling protocol with SH06 as an example. b The spatial locations of tumor sectors for SH06 (most left panel), the linear relationship between the physical distance and genetic distance (middle left panel, "Methods"), the linear relationship between the physical distance and transcriptomic distance (middle right panel, "Methods"), the linear relationship between genetic distance and transcriptomic distance (most right panel, "Methods"). 95% confidence relationship is plotted as the shaded area in pink. c The linear relationship between the physical distance and genetic distance across all patients. d The linear relationship between the physical distance and transcriptomic distance across all patients. e The linear relationship between genetic distance and transcriptomic distance across all patients. Source data are provided as a Source Data file.

In order to survey the genotypic and phenotypic heterogeneity, whole exome sequencing (WES, mean coverage of 98x) and RNA-seq were carried out for all samples ("Methods", Supplementary Data 2). With somatic variants called across all the samples, we found that *CTNNB1*, *TP53*, and chromosomal 1q amplifications are common truncal events shared across sectors (Supplementary Fig. 2a, c). Among all the patients, a wide range of tumor mutation burden (TMB) was found (1.5–11.0 perMb) and tumors with high TMB tend to enrich for mutational signatures such as aristolochic acid (AA) signature (SBS22, Supplementary Fig. 2b). In general, these patients had a similar overall genomic profile as several earlier studies including TCGA[29] and the PLANET cohort[30] (e.g., Supplementary Fig. 2d and Supplementary Data 2).

## Isolation-by-distance relationship across the genotypic and phenotypic space

Multiple hypotheses have been raised regarding the spatial organization of tumor heterogeneity[6,23]. For example, in colorectal cancer (CRC), previous studies found that subclones tend to be well organized in geographically distinct regions in pre-cancerous lesions. However, this clear segregation will rapidly progress to a spatially variegated distribution in carcinoma[23], suggesting that spatial mixing might be a hallmark of cancer. In order to survey the sITH, we measured genetic differentiation between multiple tumor populations (sectors) based on the genetic distance which characterizes the proportion of private mutations between two sectors ("Methods"). When we correlated the genetic distances with the physical distances of tumor sectors, a wide range of correlations were found across the patients. For example, SH06 has very strong genetic differentiation across sectors and shows a strong positive relationship between genetic differentiation and the physical distance (Fig. 1b). In evolutionary genetics, positive correlation between physical and genetic distance is known as the isolation-by-distance (IBD) relationship[31]. In addition to overall genetic differentiation, when we performed clonal deconvolution of multiple tumor sectors and measured distances in clonal composition between tumor sectors ("Methods"), we also observed the IBD relationship consistently (Supplementary Figs. 3 and 4). Thus, the IBD relationship is robust across multiple distance metrics (Supplementary Note 1 and Supplementary Figs. 3 and 4). Throughout the patient cohort, most patients show strong IBD pattern among the sectors (Fig. 1c).

Parallel to the genetic differentiation, when we surveyed the phenotypic (transcriptomic) differentiation based on the dissimilarity across tumor sectors ("Methods"), strong IBD relationships were also found across patients (Fig. 1b, d) and were robust to many confounding factors such as tumor purity and copy number alterations (Supplementary Note 2 and Supplementary Figs. 5–7). Correlating the genetic differentiation with the transcriptomic difference, we found that large proportions of phenotypic variation can be explained by genetic differences (Fig. 1b, e). This suggests that genetic changes, even though vary greatly across tumors, can have a very strong impact on the phenotypic evolution (Fig. 1e). Taken together, we observed an IBD pattern where physically closer sectors are genotypically and phenotypically more similar across all the tumors[32,33].

## Spatially variegated blocks in HCC

The IBD distribution provides an intuitive overview of sITH along one dimension. However, the spatial distribution of sITH within solid tumors is significantly under-explored. The phylogeographic sampling provides a comprehensive means surveying the spatial heterogeneity across the tumors. Focusing on the 10 patients with at least 10 sectors (mean = 21 sectors per tumor), we found that even though IBD is consistent across patients, many patients have a distorted linear relationship. For example, two parallel linear relationships exist in patient SH05 and many nearby samples have surprisingly high genetic differentiation

(top left corner of Fig. 2a). In evolutionary genetics, distorted IBD relationship is often driven by uneven spatial distribution[34]. By calculating local Geary's C, a spatial statistic that measures the local spatial autocorrelation[35], we found multiple regions within the tumor have a much lower correlation between genetic information and spatial location (Supplementary Fig. 8). By applying the phylogenetic analysis, principal component analysis (PCA) as well as clonal deconvolution analysis to patient SH05 ("Methods", Supplementary Note 3 and Supplementary Fig. 9), we indeed discovered two major clades of tumor sectors clearly separating the samples into two groups (Fig. 2b, c). Thus, the strong differentiation between subgroups within SH05 generates strong discontinuity in the IBD relationship.

In order to systematically explore the overall landscape of spatial heterogeneity across the 10 patients, we clustered the tumor sectors based on the phylogenetic distance between the samples ("Methods"). To choose the optimal number of clusters and compare the spatial segregation of samples between patients, we calculated the Calinski-Harabasz (CH) Index which measures the ratio of genetic differences (i.e., phylogenetic distance) between clusters vs within clusters ("Methods"). Using the CH index, we found the optimal number of clusters are often between two to three (e.g., SH05, Fig. 2e), even though higher numbers of clusters are found for a few patients (e.g., SH03, Fig. 2f). Interestingly, when we inspected the geographic distribution of these clusters, samples from the same cluster often locate in spatially continuous regions within the tumor, partitioning the tumors into variegated blocks (Fig. 2e–g and Supplementary Fig. 3). Thus, we referred "spatially distinct clusters" that partitioning the tumor as "spatial blocks" of the cancer. Strikingly, the genetic differentiation between blocks is often much higher than genetic differentiation within blocks (Fig. 2d), and regions near the block boundary tend to have low spatial correlation (Supplementary Fig. 8). Thus, the heterogeneity within HCC is spatially variegated with regional homogeneity within blocks, but large differentiation between blocks. Parallel to the natural world, this spatial distribution is very similar to our planet where tectonic plates partition the surface of the earth into discrete regions[36]. For example, two blocks exist in patient SH05 and a central ridge of high regional genetic diversity ($\theta_R$) was found at the interface of two blocks in SH05 ("Methods", Fig. 2e and Supplementary Fig. 10). Across the patients, the spatial blocks had a wide variety of shapes (Fig. 2e–g and Supplementary Fig. 3). Taken together, we found that HCC has block-shaped spatial heterogeneity across the tumors (Fig. 2g and Supplementary Fig. 3).

## Block-shaped phenotypic heterogeneity mirrors the genotypic heterogeneity

As tumors often have block-shaped heterogeneity at the genetic level, we wondered how the transcriptomic heterogeneity might co-evolve spatially within a tumor. Using SH05 as an example, when we analyzed the transcriptomic divergence (Fig. 2h), the phylogenetic relationship (Fig. 2i) as well as the PCA map (Fig. 2j), we found similar block-shaped transcriptomic heterogeneity largely matching the corresponding pattern at the genetic level. When we clustered the transcriptomic profiles into optimal number of clusters, we observed similar spatial blocks between the phenotypic and genotypic level, even though differences do exist in several patients such as SH11 and SH13 (Supplementary Fig. 3). Taken together, phenotypic heterogeneity largely mirrors the genotypic heterogeneity with similar spatial pattern across the tumors.

## Virtual micro-dissection of bulk transcriptome identified four HCC subtypes

The block-shaped spatial heterogeneity indicates non-gradual sITH in HCC and the spatial heterogeneity is driven by the geographic distribution of multiple cell types including tumor, stroma, and immune cells. In order to understand biological processes and cellular

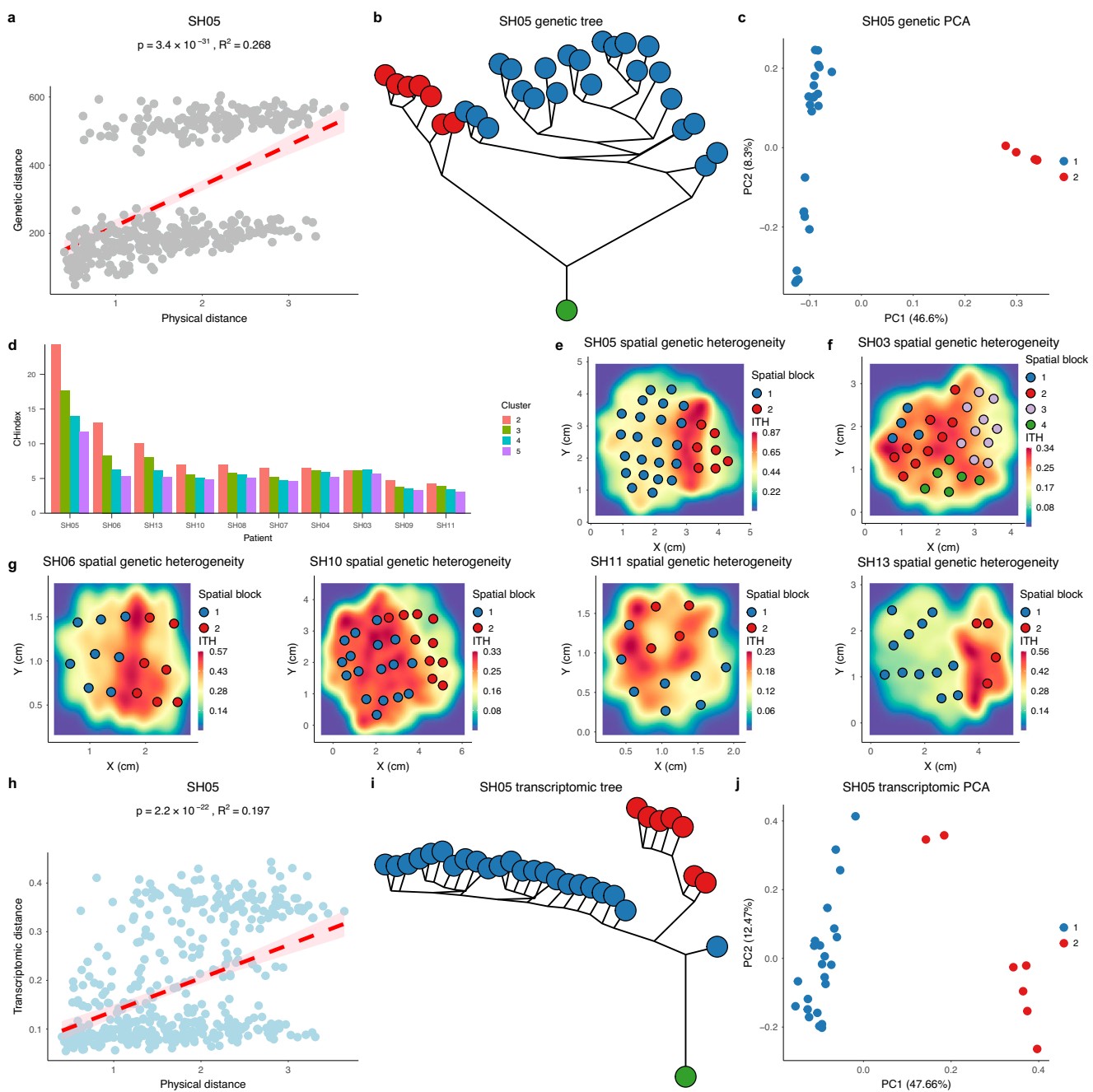

**Fig. 2 | Spatially variegated blocks in HCC. a** Genetic distance vs the physical distance relationship for SH05. Two linear relationships were found in SH05. 95% confidence relationship is plotted as the shaded area in pink. **b** The maximum parsimony tree of SH05 based on the genetic changes. **c** The PCA map of SH05 based on the genetic changes. **d** The CH index calculated for different clusters ($k = 2-5$) across the 10 patients with at least 5 sectors. **e** The spatial (regional) heterogeneity across the 2D space for SH05. The heatmap indicates the level of regional ITH ($\theta_R$). **f** The spatial (regional) heterogeneity across 2D space for SH03.

**g** The spatial (regional) heterogeneity across 2D space for the other four patients (SH06, SH10, SH11 and SH13) with SPH. The heatmap indicates the level of regional ITH ($\theta_R$). **h** Transcriptomic distance vs the physical distance plot for SH05 with the same coloring schema as panel a. 95% confidence relationship is plotted as the shaded area in pink. **i** The neighbor joining tree inferred using the transcriptomic distance for SH05. **j** The PCA map of SH05 based on the transcriptomic profile. Source data are provided as a Source Data file.

components driving the transcriptomic heterogeneity in HCC, we employed a reference-free deconvolution method[37] to deconvolute the bulk transcriptome into subcomponents (known as compartments)[38]. In order to capture as much transcriptomic heterogeneity as possible for HCC, we used the TCGA-LIHC cohort ($n = 369$) as a reference set and identified three compartments positively correlated with tumor purity (i.e., phenotypes associated with tumor cells) and one compartment negatively correlated with tumor purity reflecting the microenvironmental changes (Supplementary Fig. 11).

Using functional enrichment analysis of factor genes associated with each compartment[38], we found that compartments positively correlated with tumor purity are enriched for metabolism, cell cycle, Wnt pathway respectively, while the compartment negatively correlated with tumor purity is enriched for extra-cellular matrix (ECM) function (Fig. 3a and Supplementary Fig. 11). Across the cohort, downregulated metabolism and increased cell cycle are strongly associated with disease progression[39] and patient survival in HCC (Fig. 3b–d and Supplementary Fig. 11).

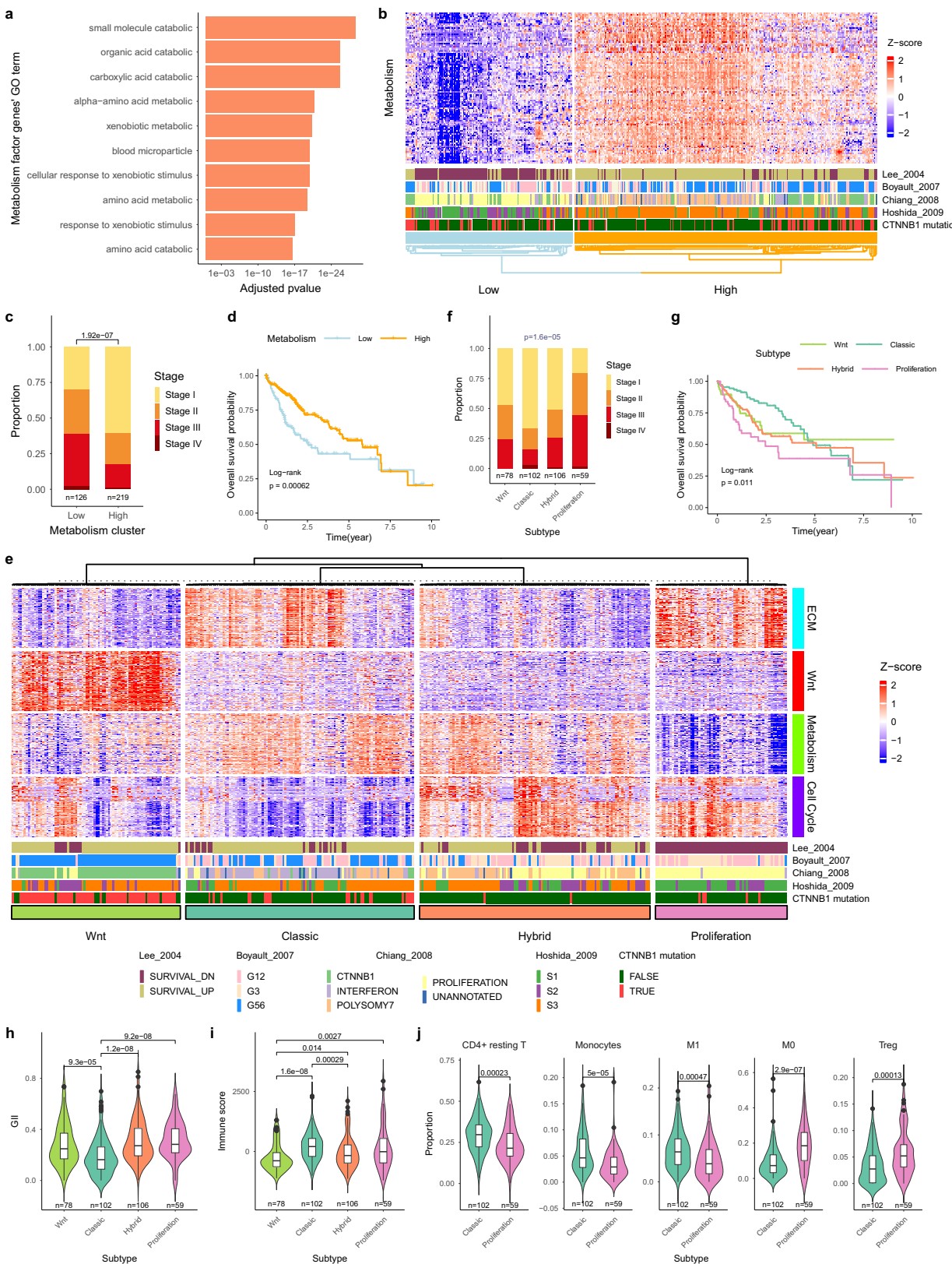

Combining all four compartments, we discovered four distinct transcriptomic subtypes for HCC (Fig. 3e, "Methods"). The first group is enriched for *CTNNB1* mutations and has strong activation of the Wnt pathway. Since this subgroup significantly overlaps with the *CTNNB1* subtype from Chiang et al.[40], we thus name this subgroup as the Wnt subtype. Among the other three subgroups, one subtype had molecular phenotypes closest to the normal with low levels of cell cycle,

high expression of metabolic pathways, and activated ECM. As patients from this group are enriched for early-stage tumors with good survival (Fig. 3f, g), we name it as the classic subtype. We also identified one subtype with the most advanced phenotypes including high cycle cell and decreased metabolic functions. Since patients in this subtype significantly match the proliferation subtype from Chiang et al.[40] and are enriched for late-stage patients (Fig. 3e–g), we name it as the

**Fig. 3 | Virtual deconvolution identifies four HCC subtypes. a** The GO enrichment for factor genes from the metabolism compartment (adjusted *p*-value, "Methods"). **b** Clustering based on factor genes from the metabolism compartment identified metabolically high and low subgroups. **c** The clinical stages of metabolically high and low subgroups. The *p*-value was calculated using Fisher's exact test. **d** Survival plot for the metabolically high and low subgroups. **e** Clustering based on four compartments identified four distinctive subtypes known as Wnt, classical, hybrid, and proliferation subtypes for the LIHC cohort. The literature known subtypes as well as *CTNNB1* mutations are marked below the patients. **f** The clinical stage of the four transcriptomic subtypes. The p-value was calculated using chi-squared test. **g** Survival plot for the four transcriptomic subtypes. **h** Boxplot of Genome Instability Index (GII) distributions across four subtypes. The lower and upper hinges represent the 25th and 75th percentiles, respectively. Whiskers extend up to 1.5 * IQR (inter-quartile range) from the hinges. Violin outlines illustrate probability density, with width proportional to data density in that region. **i** Boxplot of immune score distribution across four subtypes (the same boxplot style as Fig. 3h). **j** Boxplot of immune cell differences between classic and proliferation subtypes (the same boxplot style as Fig. 3h). Source data are provided as a Source Data file.

proliferation subtype. The other subtype has phenotypes in between the classic and proliferation subtype and was named the hybrid (i.e., intermediate) subtype[38]. Across classic, hybrid, and proliferation subtypes, we observed an increasingly higher proportion of late-stage patients with worse survival (Fig. 3f, g).

In order to explore genomic changes differentiating these subtypes, we systematically compare the tumor and non-tumor components across subtypes. As the Wnt subtype is strongly associated with *CTNNB1* mutations and represents a unique subclass of tumors, we focused on differences between the other three subtypes. When we compare genetic changes in the tumor cells, we noticed that a strong increase in genome instability between the classic and the other two subtypes, accompanying the upregulation of cell cycle related pathways (Fig. 3h, *p*-value < $10^{-6}$). When we compared the non-tumor components, in particular the immune cell composition ("Methods"), the hybrid subtype has an immune-exclusive status (Fig. 3e, i and Supplementary Fig. 12), while the proliferation subtype is immunologically exhausted with higher M0 macrophages and Tregs (Fig. 3j). Taken together, a collection of genotypic and phenotypic changes is associated with subtype differences in HCC (Supplementary Fig. 13).

### Multiple transcriptomic subtypes spatially coexist within a tumor

With the virtual microdissection approach, we have systematically explored the transcriptomic heterogeneity between patients. Within patient tumors, we discovered block-shaped spatial heterogeneity with large phenotypic and genotypic differences between spatial blocks. We wondered how this sITH within patients might be linked to inter patient differences. By clustering all our samples with the TCGA patients and assigning subtypes to the samples (Supplementary Fig. 14), four patients (SH05, SH06, SH10, and SH12) were found to have mixed transcriptomic subtypes within the same tumor. SH05 and SH10 have mixtures of hybrid and proliferation subtypes (Fig. 4a–c), while SH06 has a mixture of classic and proliferation subtypes (Fig. 4d). The spatial partition of subtypes matches almost perfectly with spatial blocks at the transcriptomic level (Fig. 4a, c, d and Supplementary Fig. 3). When we performed differential expression analysis between different spatial blocks within a tumor, we found that transcriptomic differences within patients are highly concordant with differences between transcriptomic subtypes at the inter-patient level (Fig. 4b, e, i). Inspecting genetic differences between different spatial blocks in the patients with mixed transcriptomic subtypes, we found that putative oncogenic drivers are often associated with the spatial blocks with more aggressive phenotypes. One particularly striking observation is a significant increase in chromosomal copy number alterations (CNA) in SH06 as the sectors evolved from the classic subtype to the proliferation subtype (Fig. 4f, g), matching the differences in CNAs observed at the cohort level (Fig. 3h). Even though large chromosomal changes have been repeatedly found driving punctuated evolution in tumor initiation[16–19], the observation here suggests that large chromosomal changes might also be driving tumor progression see Discussion. The fourth patient (SH12) had only five sectors and was not included in the 10 patients surveying sITH. A mixture of

the Wnt and classic subtypes exist in SH12, correlating with a subclonal *CTNNB1* mutation (Fig. 4h, i).

### The role of Darwinian selection in shaping the spatial heterogeneity

The spatially co-existing transcriptomic subtypes in different geographic locations of the tumor is very similar to an era in ancient Chinese history known as the Warring States (Fig. 5a)[41]. During the Warring States period (475–221 BCE), seven different states coexisted and competed with one another until the rise of the Qin dynasty which unified China into the first empire. Thus, we name the spatial distribution as the spatially competing distribution (SCD). As tumors can only start with one transcriptomic subtype, the coexistence of multiple subtypes primed a fundamental question: what are the evolutionary forces driving the origin of the SCD? From the subtype comparison, we observed that tumor cells with more aggressive subtypes often have additional genetic changes (e.g., much higher genome instability, Fig. 3h) and advanced phenotypes (e.g., higher cell cycle, Fig. 3e), representing derived populations further evolved from the benign subtype during the late stage of tumor progression. Thus, the spatially competing distribution might represent a critical stage of disease progression where phenotypically aggressive subtypes have emerged and are in the process of replacing relatively benign groups (Fig. 5a).

One prediction from the neutral theory is that the size of a clone depends on the time of origin and we would expect later-arising clones to be in minor frequencies under neutrality[42]. However, we found the size of the derived (i.e., aggressive) subtypes are often much larger than the ancestral (i.e., benign) subtypes, contrasting the prediction from the neutral evolution. For example, in patient SH05, the tumor region with less aggressive phenotypes (i.e the hybrid subtype) is much smaller (*n* = 5 sectors) than the area with the aggressive phenotype (i.e., the proliferation subtype, *n* = 25 sectors, Fig. 5a). Using the classical imbalance index in phylogenetics known as the Sackin index S, we also found significant evidence for clonal asymmetry for SH05 (*p*-value < $10^{-4}$, Fig. 5b). In all three patients in the SCD, more aggressive transcriptomic subtypes always occupy larger proportions of the tumors (i.e., clonal asymmetry, Fig. 5b), rejecting neutrality for these patients. In order to understand the prevalence and significance of this observation, when we extended the analysis to the PLANET cohort[30], we observed clonal asymmetry for all three additional patients in the SC distribution where phenotypically derived clones are consistently larger than ancestral clones (Fig. 5b).

Parallel to the clonal asymmetry at the phenotypic level, the other evidence for ongoing positive selection at the genetic level was the existence of subclones at intermediate frequencies. Using MOBSTER[43], we detected subclonal selection in all tumors with the SC distribution (Fig. 5c, d and Supplementary Fig. 15), which suggests that Darwinian selection can be the major evolutionary force driving tumor evolution in the SC distribution. Interestingly, when we extended the same statistical tests to the full cohort, we found varying levels of non-neutral evolution across the cohort, suggesting that Darwinian selection together with several other evolutionary forces (e.g., geographic isolation) can contribute to the block-shaped spatial heterogeneity (see Discussions).

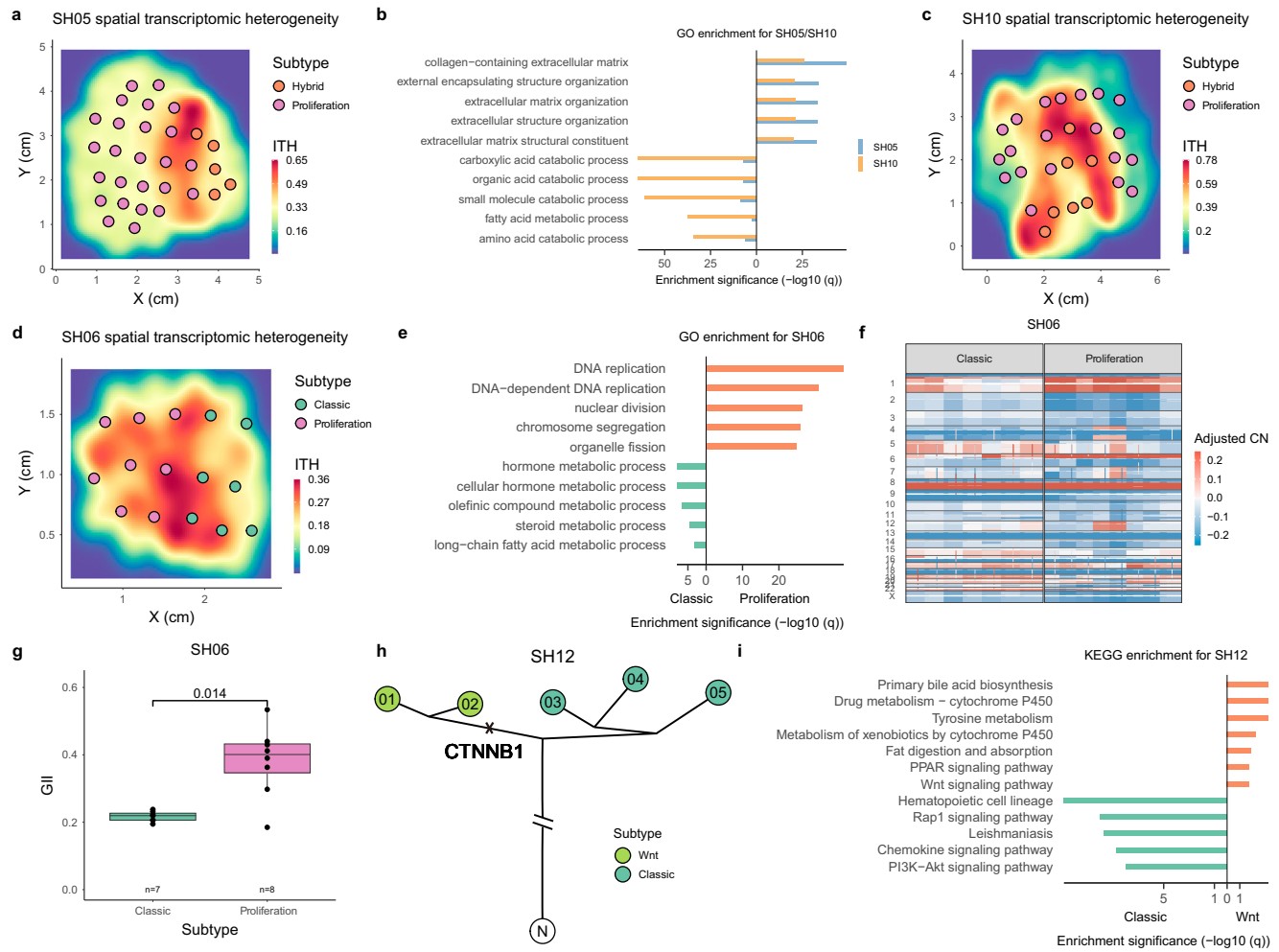

**Fig. 4 | Multiple transcriptomic subtypes within tumors. a** Multiple transcriptomic subtypes in patient SH05 with colors of the sectors representing different transcriptomic subtypes. The background color illustrated the regional heterogeneity ($\theta_R$). **b** GO enrichment results for differentially expressed genes between the two subgroups of transcriptomic subtypes in SH05 and SH10 (adjusted *p*-value, "Methods"). **c** Multiple transcriptomic subtypes in patient SH10. **d** Multiple transcriptomic subtypes in patient SH06. **e** GO enrichment results for differentially expressed genes between the two subgroups of transcriptomic subtypes in SH06 (adjusted *p*-value, "Methods"). **f** The genome wide copy number profiles for the sectors from the classic and proliferation subtypes in SH06. **g** The boxplot for the GII values for the sectors from the classical and proliferation subtype in SH06. The lower and upper hinges represent the 25th and 75th percentiles, respectively. Whiskers extend up to 1.5* IQR from the hinges. **h** The phylogenetic tree of SH12 with *CTNNB1* mutation being a subclonal mutation. **i** KEGG enrichment results for differentially expressed genes between the two subgroups of transcriptomic subtypes in SH12. Source data are provided as a Source Data file.

## sITH can significantly confound patient diagnosis and treatment

The extensive sITH raised an important question: how this spatial heterogeneity might affect clinical practice. From the literature, we curated a list of diagnostic and treatment-related biomarkers, including a transcriptomic predictor for the micro-vascular invasion (MVI) and immunotherapy biomarkers such as the GEP score (pan cancer) and the inflammatory score (HCC) (Supplementary Data 3). By plotting the variation of these biomarkers, we discovered extensive heterogeneity between and within patients (Fig. 6a and Supplementary Fig. 16). Comparing the variance of these biomarkers across patients (Fig. 6b), we found that a few patients such as SH06, SH08, and SH09 tend to have higher variance in these biomarkers. In general, the variance of these biomarkers is rather heterogeneous across patients.

In addition to intra-tumor heterogeneity, when we look at the distribution of these biomarkers across transcriptomic subtypes (Fig. 6c), we found that some biomarkers correlate strongly with disease progression. For example, the MVI score has increasingly higher values from the classic to the proliferation subtype. On the other hand, the TIDE score had relatively similar values across subtypes, suggesting that different biomarkers can have different trajectories with disease progression. When we dissected the spatial distribution of these biomarkers across patients, we found that some biomarkers had highly concordant spatial distribution with spatial blocks (e.g., the GEP score, Fig. 6d), while others have a heterogeneous spatial distribution irrespective of the spatial blocks (e.g., Cytolytic scores, Fig. 6e and Supplementary Figs. 17 and 18). This suggests that spatial segregation of biomarkers across the tumor might not always follow the overall sITH. Taken together, sITH can significantly confound patient clinical diagnosis and treatment and a single biopsy is not enough for clinical decisions (see Discussion).

## Discussion

Using a phylogeographic approach and one of the largest datasets for studying sITH[7,21], we have systematically dissected the genotypic and phenotypic spatial heterogeneity across a large number of tumor sectors from 13 HCC patients. Contrary to the spatial mixing observed in CRC, spatial heterogeneity in HCC mirrors the geography with an isolation-by-distance pattern. The spatial heterogeneity in HCC is organized into spatial blocks with large genotypic and phenotypic divergence. Using a virtual dissection approach, we discovered a

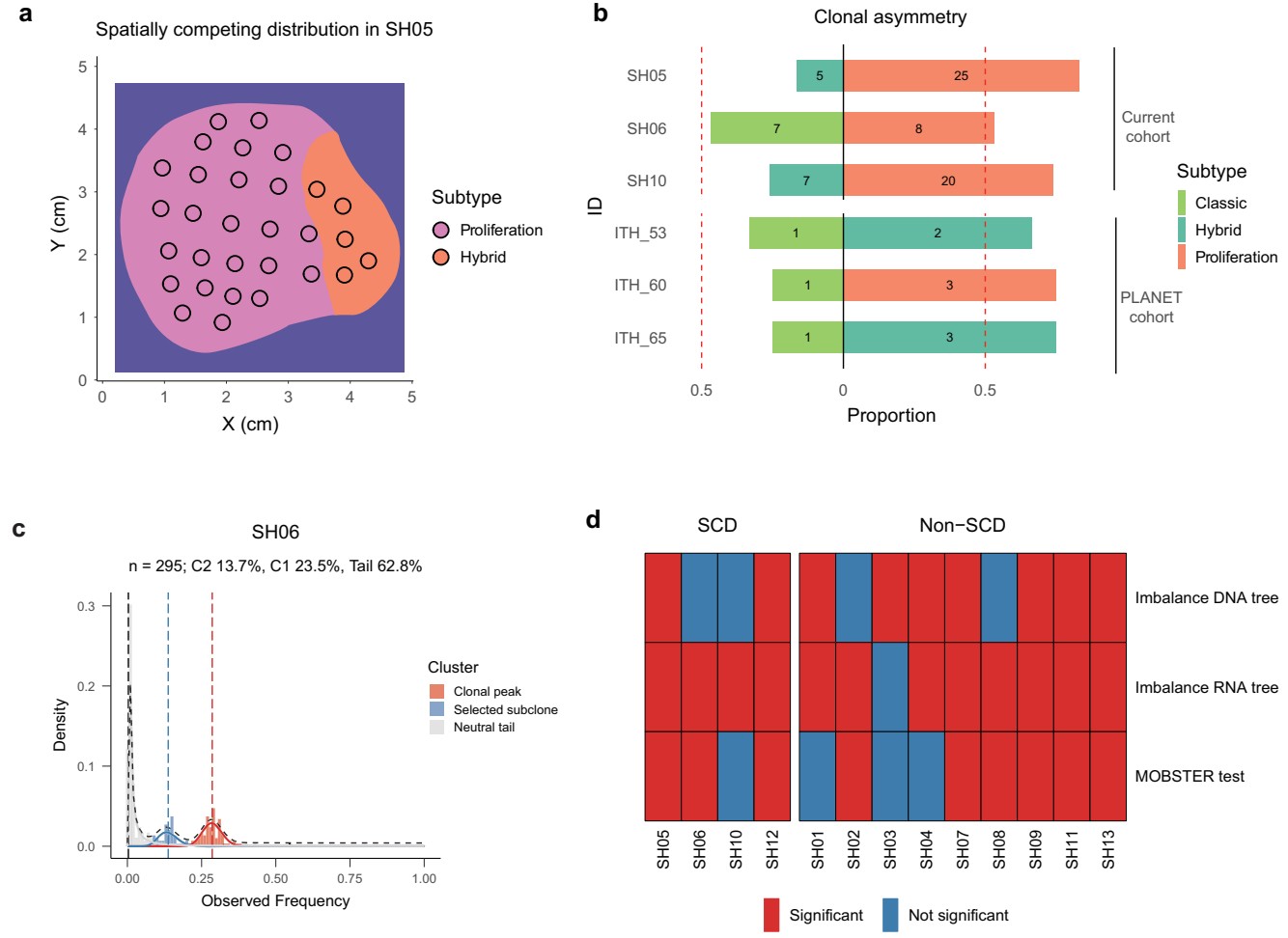

**Fig. 5 | Darwinian selection in patients with the spatially competing distribution. a** In the SC distribution, clones with more advanced phenotypes often take a wider range distribution (e.g., SH05). **b** The spatial proportions of different transcriptomic subtypes in patients with SCD from the current study and the PLANET study[30]. The proportion as well as the number of sectors in different subtypes were shown in the horizontal bars. Colors mark different transcriptomic subtypes. **c** MOBSTER output for patient SH06, a selective subclone (marked as blue) together with the neutral tail (gray) and a clonal peak (in red) are shown. **d** Heatmap of significance across multiple neutrality tests across the patients. Testing clonal asymmetry in DNA trees (Imbalance DNA tree) and RNA trees (Imbalance RNA tree) are tests of tree shapes using the Sackin S statistic. Source data are provided as a Source Data file.

spatial distribution defined as the spatially competing distribution where spatial blocks within a tumor have different transcriptomic subtypes in 30% of patients. The spatially competing distribution represents a critical transitional stage where multiple phenotypically distinct subpopulations coexist and compete during disease progression. By integrating the spatial distribution and clonal composition, we found that Darwinian selection is the driving force for disease progression in patients with SC distribution and can also contribute to the block-shaped sITH. Taken together, we have generated one of the largest datasets for studying macroscopic sITH and unraveled a dynamic landscape of sITH.

The study of sITH provides several insights into tumor evolution. First, limited sampling of sITH commonly practiced in the field might lead to the "blind man and an elephant" bias. For example, we can imagine different mode of tumor evolution can be inferred if we take samples within (linear and neutral evolution) or between different spatial blocks (branched and non-neural evolution). The extensive spatial sampling provides a holistic overview of the whole tumor and different modes of tumor evolution observed earlier might be confounded by sITH. Secondly, even though we found that natural selection is an important factor driving the spatial heterogeneity in the patients with SC distribution, they are exceptional cases with both

large phenotypic and genotypic differences. It remains unknown what are the evolutionary forces driving the collective pattern of spatial heterogeneity including block-shaped heterogeneity and clonal asymmetry. Even though varying levels of selective signal across the patient cohort were detected, previous studies in evolutionary genetics suggest that geographic isolation together with a suite of non-selective forces can also contribute to the spatial pattern[44]. Taken together, this study pointed directions how we can dissect the evolutionary forces driving the spatial heterogeneity and how we may formulate more rigorous methods in detecting natural selection at both genotypic and phenotypic levels (Supplementary Note 4).

The study of spatial heterogeneity also highlighted challenges facing patient diagnosis and treatment. Even though truncal driver mutations can be well represented by a single biopsy[45], many of the diagnostic and treatment biomarkers are phenotypic measurements and often have extensive spatial variation. For example, the immune microenvironment tends to have high diversity and does not always follow the genetic diversity observed in tumor cells (Supplementary Fig. 18). Moreover, when we compare the biomarker distribution between the central and peripheral regions of the tumors, no significant differences were found except the vascular invasion (Supplementary Fig. 19). As we found earlier, ancestral clones can reside in

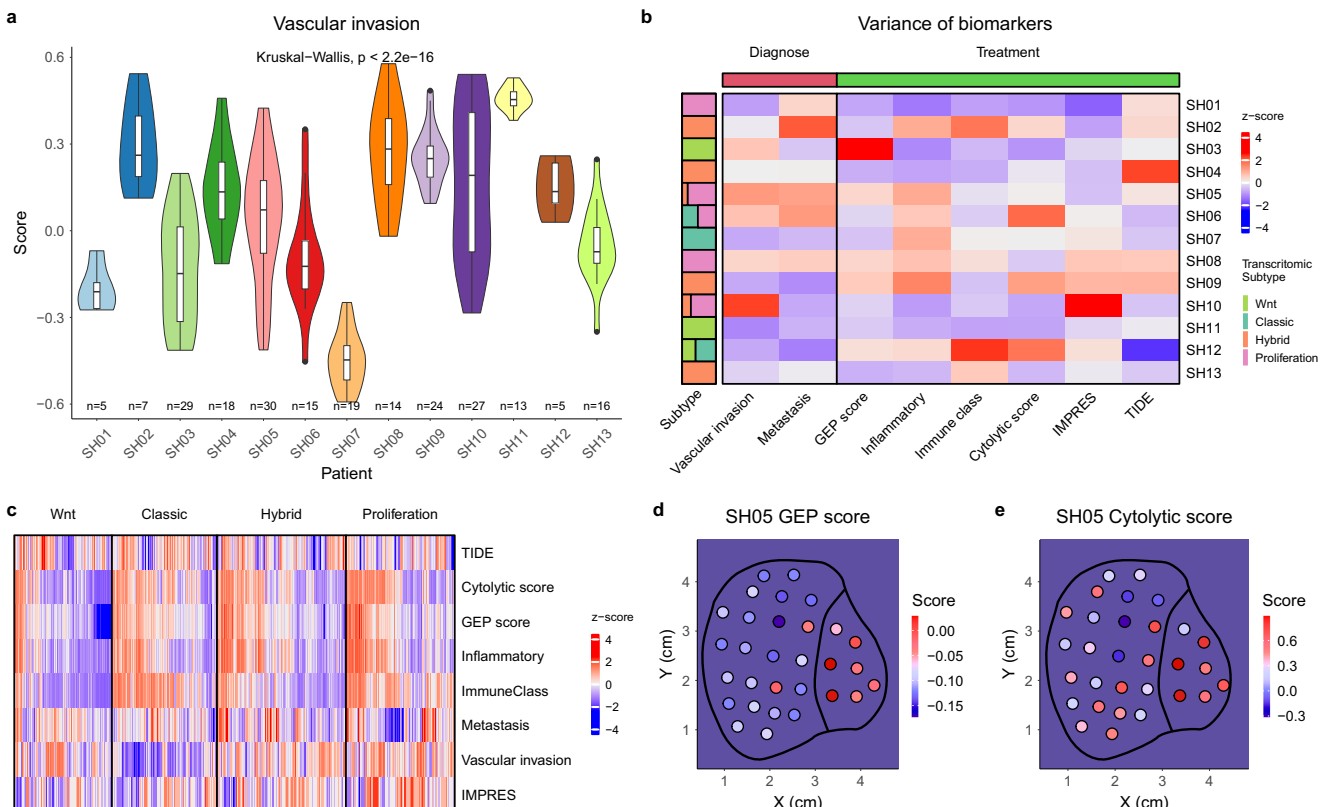

**Fig. 6 | Spatial heterogeneity of biomarkers. a** Boxplot showing within and between patient variability for the vascular invasion (Kruskal Wallis two-sided test, For the style of the boxplot plot, see "statistical analysis" section in "Methods"). **b** The variance of different biomarkers (row) in different patients (column). The left

bar indicates whether the patient has multiple transcriptomic subtypes. **c** Heatmap for biomarkers across transcriptomic subtypes. **d** The spatial distribution for the GEP score in SH05. **e** The spatial distribution for the cytolytic score in SH05. Source data are provided as a Source Data file.

different parts of the tumors and tumor growth can take a wide variety of shapes which might yield complex spatial distribution without systematic differences between inside and outside of the tumor. Thus, how to design spatially guided approaches for patient diagnosis and treatment can be a promising direction for the community.

## Methods

### Patient recruitment and spatial sampling
From the Ditan Hospital affiliated with Capital Medical University in Beijing, we recruited 13 treatment naïve HCC patients with approval from the Institutional Review Board (IRB) of the Ditan Hospital. Signed informed consent was obtained from each patient before surgery. After surgical operation, a single slice of tumor sample was harvested from the center of the tumor and multiple tumor sectors (3 mm or 5 mm in diameter) were harvested across the 2-D space depending on the size of the tumor (Fig. 1a). In total, 222 tumor sectors ranging from 5–30 regions per tumor (T) and 13 matched adjacent non-tumor liver tissues (N) were harvested (Supplementary Fig. 1). The physical coordinates of the tumor sectors were subsequently extracted based on the sampling image. Sex and/or gender were not considered in the study design.

### Genomic sequencing and variant calling
DNA and RNA were extracted from tissue samples using AllPrep DNA/RNA Mini Kit(Qiagen). Exome libraries were constructed using the Agilent SureSelect Human All Exon V6. RNA sequencing libraries were generated using NEBNext® UltraTM RNA Library Prep Kit from Illumina® (NEB, USA) following the manufacturer's recommendations. Both exome and RNA sequencing were conducted at Novogene Co., Ltd. with 150 bp paired-end reads.

After quality control which includes trimming adapters and removing low-quality reads, we mapped short reads to the hg19 reference genome using BWA MEM (v0.7.17)[46]. Duplicated reads were subsequently marked using MarkDuplicates and base quality scores were recalibrated using GATK4 (v4.1.3.0)[47]. Somatic variants were called using Mutect2[48]. Variants were further filtered using GATK FilterMutectCalls with default parameters and were annotated with Oncotator (v1.9.9.0)[49]. Somatic copy number variations as well as tumor purity were inferred using Sequenza (v3.0.0)[50]. The genome instability index (GII) is defined as the proportion of the genome with abs(log2(CN/2)) > 0.2, where CN is the copy number of the focal genomic region[51].

### Signature analysis
We used SigProfiler[52] to perform signature analysis. We first used SigProfilerMatrixGenerator[53] (v1.2.1) to create a mutation matrix for all types of mutations. We then applied SigProfilerExtractor (v1.1.4) to extract de-novo mutational signatures from the mutational matrix and further deconvolute de-novo signatures to the COSMIC signatures (v3.2)[52].

### Curating putative driver events in HCC
A list of 62 driver genes for HCC was extracted from a recent integrative analysis of HCC genomes[54]. Significantly mutated copy number alterations were extracted by reanalyzing the TCGA-LIHC dataset using GISTIC2 (v2.0.23) algorithm[55] (Supplementary Fig. 2a, d).

### Transcriptomic analysis
The raw RNA sequencing reads were aligned to the reference genome using STAR (v2.7.2b)[56] and further quantified using RSEM (v1.3.1)[57]. We

used DEseq2 (v1.30.1)[58] to normalize expression levels of different samples and performed differential expression analysis. Transcriptomic distance between two samples was calculated as the (1-Spearman correlation) between two samples using top most variable 3000 genes (based on median absolute deviation, MAD). In order to explore the influence of tumor purity on tumor spatial phenotypic heterogeneity, we identified genes that were positively correlated with tumor purity in our data as those with Spearman correlation coefficient greater than 0.5 and used this subset of genes to calculated the transcriptomic distance (Supplementary Note 2). In order to explore the influence of copy number variation on tumor phenotypic spatial heterogeneity, we constrained the gene list to genes with no copy number variation in all sectors of a given patient and calculated transcriptomic distances based on this subset of genes (Supplementary Note 2).

### Phylogenetic, clonal deconvolutional, and population genetic analysis of tumor samples

The phylogenetic and clonality analysis followed similar steps as our previous work[30]. The genetic distance between two sectors was calculated as the Hamming distance between the two mutation lists called from two sectors. We used the Analysis of Phylogenetics and Evolution (APE) from R to perform the phylogenetic analysis. We first tabulated the presence and absence of mutations across sectors for each patient, maximum parsimony[59] was employed to construct the phylogenetic relationship. In addition to the maximum parsimony, Neighbor-joining algorithm[60] was also employed to construct the phylogenetic relationship based on the Hamming distance calculated using the presence and absence of mutations. We also used the 1-Spearman correlation as the distance at the transcriptomic level to construct the neighbor-joining tree at the transcriptomic (RNA) level.

Based on somatic variants and local copy number variations, PyClone (v1.13.1)[61] was employed to dissect the clonal composition of multiple tumor samples, which infers the predicted number of clones, clonal structure, mutation assignment, and mutation VAF distribution in each clone. Before PyClone analysis, we have performed these steps: 1) take a union of all the SNVs across all sectors for a given patient; 2) extract the number of reference and alternative allele count at the SNV positions for all sectors (based on Samtools mpileup). 3) We required the SNVs to have a frequency > 0.05 in at least one sector (to filter away possible false positives). Clonality inference was performed based on SNVs only as variant frequency estimation for indels are much less accurate.

Similar to our previous study[30], CITUP (v0.1.0)[62] was used to construct the optimal clonal phylogeny from the clonal composition inferred from PyClone. We first filtered away clones with only one mutation from the PyClone output and subsequently took the top 10 clones with the highest number of mutations[30] (CITUP analysis with too many clones can be computationally too intensive[62]). Clonal phylogenies were visualized using the ggplot2 package in R.

### Alternative approaches correlating the genetic distance and physical distance

In addition to correlating the genetic distance and physical distance, we also explored alternative approaches. In particular, when we correlated the phylogenetic distance as well as the clonal distance with physical distance, we observed a consistent linear relationship (i.e., IBD relationship, Supplementary Note 1). The phylogenetic distance was extracted from the phylogenetic relationship. The clonal distance is measured as the differences in the clonal composition and is calculated as the Euclidian distance between samples based on their clonal composition. To be more precise, if there are n clones presented in two samples with proportions ($p_1$, $p_2$...$p_n$) and ($q_1$, $q_2$, $q_3$...$q_n$), we calculated the distance in the clonal composition between two samples as $\sqrt{\sum_i^n (p_i - q_i)^2}$.

### Spatial correlation and the Geary's C statistic

For each sector $i$, the local Geary's C index $C_i$ is calculated as the following formula:

$$C_i = \frac{2n^2}{\Sigma_i \Sigma_j D_{ij}^2} \sum_j w_{ij} D_{ij}^2, \tag{1}$$

where $j$ represents another sector in the tumor, and $n$ represents the total number of sectors in the patient. $w_{ij}$ denotes the spatial weight between sector $i$ and $j$. We employed the canonical form of $w_{ij}$ by taking the inverse of the physical distance between sector i and j. $D_{ij}$ represents the genetic distance between sector $i$ and $j$ which we used the Hamming distance between mutations found in sector i and j.

To test the significance of the observed value, we calculated the empirical $p$-value of the observed local Geary's C. We randomly permuted the sectors spatially and recalculated the Geary's C and computed the proportion of replicates where the permuted $C_i$s are less than the observed $C_i$. In case of neighborhood with strong spatial autocorrelation, the local Geary's C will be small because of smaller values of $D_{ij}$. On the contrary, regions with weak spatial autocorrelation will have very large $p$-values because of large values of $D_{ij}$ in the local neighborhood.

### Spatial blocks and the spatial heterogeneity of the tumor

In order to explore the spatial distribution of ITH, we clustered the genetic profiles of tumor sectors following these steps. 1) Based on the parsimony tree, we cut the phylogenetic tree into all possible k clusters for a given number of clusters k ($k = 2$–$5$). 2) In order to compare the clusters within and between clusters, we computed a statistic Calinski-Harabasz Index (CH-index)[63,64]. CH-index is commonly applied for clustering evaluation[63,64]. It is calculated as the ratio of the inter-cluster dispersion and the intra-cluster dispersion, where dispersion is measured by the mean sum of distances. For a collection of $N$ samples that are partitioned into $K$ clusters, the sum of distances within and between clusters can be calculated as:

$$SS_{within} = \sum_{k=1}^{K} \sum_{i=1}^{n_k} \sum_{j=i+1}^{n_k} \frac{d(i,j)}{n_k}, \tag{2}$$

$$SS_{between} = \sum_{i=1}^{N} \sum_{j=i+1}^{N} \frac{d(i,j)}{N} - SS_{within}, \tag{3}$$

where, $n_k$ is the number of samples in cluster $k$ ($k = 1,2,\ldots,K$), and $d(i,j)$ denotes the distance between pairwise samples $i$ and $j$. The CH-index is then calculated as:

$$CH = \frac{SS_{between} \times (N-K)}{SS_{within} \times (K-1)}. \tag{4}$$

We picked the optimal number of clusters according to the CH-index (maximum value). In the clustering of the genetic pattern, we used the phylogenetic distance as the distance metric. In the clustering of the transcriptomic profile, we use the phylogenetic distance in the transcriptomic tree as the distance metric.

In order to display the spatial heterogeneity across the 2-D space, we calculated ITH values at grid points distributed along the 2-D space (with a grid size of 2 mm in distance). We defined the local region as a circular area centered around the grid point and we have chosen the diameter as the median distance between all sectors of that tumor. The regional diversity (i.e., $\theta_R$) is defined as the average difference between all pairs of samples in the circular area. For pairs of samples, the genetic (DNA) distance is calculated as the proportion of branch mutations (i.e., non-shared mutations) among all mutations found in two samples. In order to explore how variation in sequencing depth

might affect $\theta_R$, we down-sampled the sequencing depth of all sectors to the minimum sequencing depth across all the sectors for a given patient. We found that the re-estimated $\theta_R$ values are highly similar to the original values (Supplementary Fig. 10). In order to depict the spatial variation in diversity, we generated a spatial heatmap with a coloring hue proportional to the level of $\theta_R$ using the "stat_density_2d" method from the R package ggplot2.

### Virtual deconvolution and the transcriptomic subtypes of HCC

Raw expression counts for the TCGA LIHC cohort ($n = 369$) were downloaded from the TCGA website and subsequently normalized using Deseq2[58]. We employed DECODER[37] to identify subcomponents (known as compartments) and infer compartment weights for each sample using the non-negative matrix factorization (NMF) method. We set 10,000 resampling to train the seeding matrix and DECODER identified five different compartments.

To annotate the biological functions of each compartment, we performed Gene Ontology (GO) and Kyoto Encyclopedia of Genes and Genomes (KEGG) analysis of the factor genes from each compartment using the R package "ClusterProfiler". We have chosen four compartments with significant correlation to tumor purity as the base for identifying HCC subtypes. The fifth compartment has no correlation to tumor purity and factor genes associated with the fifth compartment are not enriched for any biological function. For each compartment, we chose the top 100 factor genes from each compartment and then used the combined set of factor genes across all four compartments to perform the consensus clustering using the ConsensusClusterPlus from the R package. In the consensus clustering, we resampled 80% samples and features (expression) and hierarchical clustering was performed for each resampled dataset. We set the number of iterations to be 500 and used 1- Spearman correlation distance as the distance metric. The cumulative distribution function (CDF) curves of the consensus matrix were used to determine the optimal number of clusters. In order to assign transcriptomic subtypes to our patient samples, we combined LIHC dataset with our cohort and reperformed the clustering.

### Annotating samples for known transcriptomic subtypes

To examine whether our clustering results are consistent with known transcriptomic subtypes from the literature[39,40,65,66], we downloaded pathways associated with the transcriptomic subtypes curated in MSigDB[67] and applied GSVA analysis to each sample to compute a pathway score for each sample. Samples were then assigned to the subtype with the highest pathway score.

### Immune score and immune cell proportion

Immune scores estimated by ESTIMATE[68] were retrieved from ESTIMATE's website at https://bioinformatics.mdanderson.org/estimate/. Immune cell proportions were estimated using CIBERSORTx[69]. ConsensusTME (v0.0.1.9) was also used to infer the composition of the tumor microenvironment[70]. We calculated the immunological regional diversity in a similar fashion as the genetic and transcriptomic level. In particular, we used the dissimilarity distance (i.e., 1-Spearman correlation) between samples based on the proportion of immune cells estimated by CIBERSORTx.

### Neutrality tests

In order to test for neutrality, we performed several statistical tests: 1) To test for clonal asymmetry, we applied the Sackin test to test if the tree shapes fit the null hypothesis of equal fitness (i.e., Yule process). We used the "sackin".test from "apTreeshape" (v1.5-0.1) to perform the test[71]. 2) In order to test the existence of the selected subclone, we used MOBSTER (v0.1.1)[43] to identify the selected subclone at intermediate frequencies. In order to obtain a corrected variant allele frequency (VAF) for MOBSTER, we first used Samtools mpileup to compute the

raw VAF for all the somatic variants across all tumor sectors of a given patient. To adjust for tumor purity and local copy number, PyClone was used to infer the Cancer Cell Fraction (CCF). Sites with corrected VAF < 0.01 were removed before the subclonal deconvolution using MOBSTER.

### Biomarkers for HCC

To explore spatial heterogeneity in diagnostic and treatment-related biomarkers, we collected 8 biomarkers including TIDE[72], GEP[73] and IMPRES[74], Cytolytic[75], Immune class[76], inflammatory[77], vascular invasion[78] and metastasis[79] score. TIDE, GEP, and IMPRES score were calculated using the original method. The other biomarkers were calculated using GSVA based on their marker genes. In order to compare the biomarker distribution between the central and peripheral regions of the tumor, samples that are within r/2 from the central point of the tumor were defined as the center of the tumor, where r is the radius of the tumor.

### Statistical analysis

Linear regression was performed based on the least-square method. Multiple test correction is based on the Benjamini-Hochberg (BH) approach. When testing differences in two categories (e.g., Figs. 3h–j and 4g), the Wilcox test (two-sided test) is employed.

### Reporting summary

Further information on research design is available in the Nature Portfolio Reporting Summary linked to this article.

## Data availability

The raw data of exome and RNA sequencing have been deposited in the Genome Sequence Archive (GSA) hosted at the National Genomics Data Center (HRA002112). The TCGA-LIHC dataset was obtained from the Genomic Data Commons (GDC) data portal [https://portal.gdc.cancer.gov/projects/TCGA-LIHC]. Source data are provided in this paper.

## Code availability

The scripts for analyzing the data and plotting the results are available at GitHub hosted at https://github.com/LiuXiaodong1/HCC_SPH.

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

## Acknowledgements

We would like to thank Chung-I Wu, Rasmus Nielsen, Xionglei He, Miao Xu, Yanhua Qu and Zhengting Zou, Jianbin Chen for their constructive comments. This work is funded by the National Science Foundation of China (grant no. 32293192/32293190, 31970566, 92259303 and 32000407). L.M. is supported by the National Key R&D Program of China (2019YFA0709501) and the National Natural Science Foundation of China (11971459). W.Z. is supported in part by the Strategic Priority Research Program of the Chinese Academy of Sciences XDPB17 (XDPB17) and the National Key R&D Program of China (grant 2018YFC1406902 and 2018YFC0910400).

## Author contributions

W.Z., L.J. and L.M. co-lead the project. X.L., K.Z., Z.J., D.W., T.C. and L.J. collected the patient samples. X.L., N.A.K., S.Z. and L.M. performed the analysis. X.L. and W.Z. wrote the original manuscript with inputs from K.Z., N.A.K., Z.J., D.W., T.C., Z.L., S.Z., A.M.H., T.W., J.L., Y.S.C., Z.H., L.M. and L.J.

## Competing interests

The authors declare no competing interests.

## Additional information

¹Key Laboratory of Zoological Systematics and Evolution, Institute of Zoology, Chinese Academy of Sciences, Beijing, China. ²School of Life Sciences, Division of Life Sciences and Medicine, University of Science and Technology of China, Hefei, China. ³Department of General Surgery, Beijing Ditan Hospital, Capital Medical University, No. 8, Jingshun East Street, Chaoyang District, Beijing, P.R. China. ⁴Genome Institute of Singapore, Agency for Science, Technology and Research, Singapore, Singapore. ⁵University of the Chinese Academy of Sciences, Beijing, China. ⁶Centre for Quantitative Medicine, Program in Health Services and Systems Research, Duke-NUS Medical School, Singapore, Singapore. ⁷Institute of Pathology, Faculty of Medicine and University Hospital Cologne, University of Cologne, Cologne, Germany. ⁸School of Biological Sciences, Nanyang Technological University, Singapore, Singapore. ⁹Bioland

Laboratory (Guangzhou Regenerative Medicine and Health Guangdong Laboratory), Guangzhou, China. [10]CAS Key Laboratory of Quantitative Engineering Biology, Shenzhen Institute of Synthetic Biology, Shenzhen Institutes of Advanced Technology, Chinese Academy of Sciences, Shenzhen, China. [11]Center for Excellence in Animal Evolution and Genetics, Chinese Academy of Sciences, Kunming, China. [12]These authors contributed equally: Xiaodong Liu, Ke Zhang. [13]These authors jointly supervised this work: Liang Ma, Li Jiang, Weiwei Zhai. ✉e-mail: maliang@ioz.ac.cn; jiangli1903@163.com; weiweizhai@ioz.ac.cn

