## [Peer Review File · Nature Communications]

REVIEWER COMMENTS

Reviewer #2 (Remarks to the Author): Expert in cancer genomics, heterogeneity and evolution

The authors performed whole exome and transcriptome sequencing in 235 tumors from 13 patients with hepatocellular carcinoma in order to study the spatial intratumor heterogeneity (ITH) and how it might correlated with the clonal and transcriptional patterns across different sectors of the whole tumor.

By comparing the transcriptomic and mutation data across different tumors of the same patient, they show that half the patients exhibited gradual patterns of spatial ITH. In the other half of the patients, on the other hand, the tumors segregated into layers of clones that the authors call tectonic plates which are characterised by different transcriptomic and mutation profiles. They show that in 60% of the patients, different tectonic plates with distinct transcriptional profiles are present where the one with a more advanced (aggressive) phenotype dominates the tumor population. They use this to say that there is Darwinian selection taking place in these tumors.

This is study with a potential relevance for cancer genomics community. However there are some methodological concerns that precludes the further.

Certain aspects of the clonality analysis is not explained. See below.

1.1 Purity/ploidy estimates for these tumor are not given. Was there a cutoff? This is very basic prior to any type of clonality analysis.

1.2 What kind of CNA was performed? Didn't the authors identify any putative oncogenic CNAs?

1.3 Any mutation filtering performed prior to Pyclone analysis? Mutations present in low VAFs across many tumors might be artefactual and affect downstream clonality analysis.

1.4 Were indels and SNVs analyzed together? I would say this must be the case since the authors have exome data only. However it is not clear from the text. VAF estimates for indels are not as accurate as for SNVs. Might this affect clustering using Pyclone?

1.5 Are clone trees based on both CNAs and SNV/Indels?

1.6 Any filtering performed on clusters after Pyclone analysis? Were there clusters with predominantly indels or with SNVs/indels with low CCFs across many tumors of the patient? These might be artefactual as well.

1.7 General stats on clusters should be provided as a table (number of mutations, CCF in each sample etc)

1.8 The final clone trees in S. Fig. 3 are not annotated with putative oncogenic events. Are there subclonal drivers that characterize the patients with punctuated spatial heterogeneity?

On fixation index, sample trees and clone trees

2.1 The authors draw sample trees and clone trees for each patient (shown in panel-b and panel-k in S. Fig. 3). Sample trees are inferred from the binary mutation matrix in panel-d and show which mutations are shared across the samples. Clone trees are generated by running CITUP on the clones identified by Pyclone and would be the best approximation to the underlying subclonal structure. However all the downstream "genetic" analysis is performed on sample trees. The authors use the sample tree (binary mutation matrix) to calculate the fixation index (FST) which is then used to measure the genetic relatedness. What worries me with this is that sample trees and clone trees are not necessarily equivalent especially if the samples are polyphyletic, which is what it looks like from panel-k in S. Fig. 3. The ideal thing to do would be to compare the phylogenetic relationship between the sectors not the genetic relatedness. How would the results look like if the authors used the absence/presence of the clones instead of the absence/presence of the mutations to calculate FST?

2.2 If FST was calculated from the clonal trees then we would know if there were any specific mutations that might relate with the difference in transcriptional profile. Could the authors annotate the clone trees with clonal/subclonal putative oncogenic changes.

2.2 The authors partition the tumor along different central axes and calculate the FST-ratio, mean FST within a partition divided by mean FST across partitions. They designate patients whose tumors show fluctuations across different partitions as tectonic. I wonder if this analysis is too coarse and if the authors need a finer grid search. Some of the "non-tectonic" tumors do seem to have samples that share a substantial number of mutations (mutation heatmap in S. Fig. 3) and are indeed within physical proximity of each other (S. Fig. 1). For example

- SH03: (T01, T02, T03 T06), (T05, T07, T08, T11, T12), (T14, T15, T19, T20, T21, T22, T25, T26, T27, T29) and (T16, T17, T18, T23, T24)

- SH07: (T05, T06, T11, T12) and (T08, T14, T15, T18, T19)
- SH08: (T02, T03, T07, T08, T10) and (T04, T05, T09, T12, T13)

Other points

3.1 Please provide the citation for “genome instability index”.

3.2 On lines 333-335, the authors say all patients with Warring States distribution of clones in the transcriptomic data also have evidence of positive selection as assessed by MOBSTER. However, there are selected clones in non-Warring States patients as well (Fig. 5d). How do the authors explain this?

3.3 Throughout the text, when referring to figures please refer to which panel. For example in lines 138 and 141, it should be "Supplementary Fig. 2a" and "Supplementary Fig. 2b", respectively.

3.4 Please label panel-i S. Fig. 3 with tumor ids.

Reviewer #3 (Remarks to the Author): Expert in cancer genomics, heterogeneity and evolution

The manuscript from Liu and colleagues collects a mean of 17 samples from each of 13 hepatocellular tumours, and performs exome and RNAseq on each sample. Their analysis is based around a comparison of clonal structure (measured from DNA) and the association with gene expression. They report that tumours often segregate into a small number of large subclones, with distinct gene expression patterns, and further that one of the subclones has a putatively more aggressive gene expression pattern. The conclusion broadly is that measuring spatial heterogeneity gives a new window in cancer evolutionary dynamics, and in liver cancer in particular, that subclonal selection is common.

Spatially resolved tumour heterogeneity is interesting (and a hot topic), and the large dataset of spatially resolved DNA and RNA data is valuable. The authors push a fairly novel line of analysis (though others have looked at DNA-RNA relationships fairly extensively in cancer before), but there are many assumptions and uncertainties in the analysis that leave us feeling unsure of the strength of the conclusions.

Detailed comments:

- The FST statistic is used to measure genetic distance between populations. It wasn't clear to us how the statistic was calculated (it is not specified in the methods: some precise mathematical formulas should be given). This makes it very hard to assess the meaning of the results.

- The FST statistic requires an assessment of within-sample diversity, which is a problematic thing to measure. The methods seem to imply pyclone was used for this, but do not specify how within and between sample diversity was calculated (SNV distance between deconvolved subclones?). Further, the authors inconsistently switch between of pyclone and then MOBSTER methods to measure clone structure – why? We note that MOBSTER corrects for neutral tails: we’re not clear how these tails influence FST calculations (failing to remove them certainly risks introducing spurious clones into the FST analysis).
- We think a simpler way to measure genetic distance would be to use distances measured on phylogenetic trees. However, if phylogenetic distances are used (which we strongly recommend), then the authors need to redo their tree building, as currently they use neighbour joining which is very crude (essentially clustering) and so is not really a proper phylogenetic method. Maximum parsimony methods seem to be the accepted minimum standard in the cancer field and we suggest the authors use this (or a more sophisticated ML/Bayesian method).
- (Fig2; line 172 onwards) Because of our confusion over the FST statistic, the tectonic plates section of the manuscript was hard to follow (particularly how plates (subpopulations) were defined and what this means in terms of genetic distance). It seems the authors always divide their samples by slicing in a straight line through the centre of the tumour (line 485); why? This whole section was very unclear and difficult for us to assess.
- Again, we felt this analysis would be likely be much clearer if subpopulations were defined based on a phylogenetic analysis directly. Because evolution is always branching, there are always “subclones” so any discretisation of samples into groups is essentially arbitrary: cutting a phylogenetic tree seems to us to be a straightforward and intuitive way to make groupings.

Comments on the rest of the manuscript are made with the caveat that we were very unsure of the soundness of the genetic heterogeneity measurements in the first half of the paper:

- Is the apparent correlation between spatial heterogeneity and transcriptomic heterogeneity corrected for tumour cellularity in each sample? (fig 2h)
- How precisely is transcriptomic distance calculated? Mathematical formulas should be given.
- Is the influence of CNAs on gene expression accounted for in expression distance measurements? (there is typically a roughly linear relationship between CNAs and gene expression)
- How does immune cell composition differ across the tumours – could this be shown directly (analogous to Fig 2e-g)
- The logic that putatively aggressive clones are often larger than putatively benign clones seems reasonable evidence of positive selection. Does genetic data back this up – are there additional known drivers, or $dN/dS > 1$ in the positively selected clone? If a phylogenetic analysis is done, are the larger clones always phylogenetically newer, therefore faster growing? (relatedly, the implied association between SNV burden and aggression should be shown.)

- The tree balance statistics (Fig 5d) don't seem to show a statistically significant enrichment of signals of selection in "warring states" tumours -> the authors seem to imply this is the case? If they aren't enriched in cases where selection is (most) evident, how can we be sure that the statistics are not overcalling selection?

- We found the "tectonic plates" terminology unhelpful – couldn't these be more simply described as "subpopulations" or even "subclones"? Relatedly, tumours are 3D but tectonic plates are essentially a 2D surface, so this feels a misleading analogy. The phrase "punctuated spatial heterogeneity" is also confusing, as the word punctuated implies a temporal aspect, which we don't think is meant. Similarly the "Warring States" terminology seems unnecessary (though it is a nice metaphor): "competing subpopulations" would be simpler.

Reviewer #4 (Remarks to the Author): Expert in hepatocellular carcinoma genomics

Liu et al performed genotypic and phenotypic analysis of the spatial heterogeneity of 13 hepatocellular carcinomas in 235 spatially informed sectors in an attempt to obtain spatial distribution patterns. The authors discovered punctuated spatial heterogeneity (PSH) in 50% HCC patients and found a spatial distribution named "Warring States" by a virtual deconvolution analysis of transcriptomic profiles from TCGA LIHC cohort and the present study cohort, challenging the popular theory of neutral evolution. The extensive spatial sampling in this study provides an important resource for studying tumor evolutionary patterns at the macroscopic level and the analysis conducted are sound. However, my major concern is that the current study mainly analyzes some phenomena and lacks in-depth mechanism exploration. In addition, this study does not break through the conventional multiregional sequencing strategy, but studies more sectors of tumor specimens. In fact, spatial transcriptome sequencing can provide more information.

Comments:

1. Somatic mutations, copy number variations, gene expression matrix and PyClone results for all the samples should be included as supplementary datasets, which are essential for other researchers to reproduce the authors' findings. The mutation signature activity and mutation burden should also be listed in a supplementary table.

2. In lines 485-490, Fixation index (FST) is a well-known method in population genetics research and is based on calculating the variance of alleles in a population. However, the use of FST in this study is simplified and based on somatic variants distance (I guess) or transcriptomic distance. Authors should provide a formula to describe the details of calculating the FST in this study.

3. In lines 499-503, the authors mention how to cluster the transcriptome landscape, but miss the method of classifying tectonic plates based on spatial genetic heterogeneity. As shown in Figure 2, the details of clustering spatial genetic level are more important but unclear in Methods section.
4. It is confused about how to define the distance relationship (between or within) in Figure 2a. If I understand correctly, the classification of sample pairs (Figure 2a) is based on the clustering of spatial genetic heterogeneity (Figure 2e-g) after selecting patients with IBD (Figure 2d). Please consider reordering Figure 2.
5. Please add color legend in Figure 3c.
6. In line 267-269, the authors conclude that the hybrid subtype has an immune-exclusive status (Figure 3e and I). Please provide more evidence.
7. Please provide the comparison group for the differential expression analysis in Figure 4b, e and I. Proliferation vs Hybrid? Or Hybrid vs Proliferation? What the transcriptomic subtypes are in SH12?

Reviewer #2 (Remarks to the Author)

The authors performed whole exome and transcriptome sequencing in 235 tumors from 13 patients with hepatocellular carcinoma in order to study the spatial intratumor heterogeneity (ITH) and how it might correlated with the clonal and transcriptional patterns across different sectors of the whole tumor. By comparing the transcriptomic and mutation data across different tumors of the same patient, they show that half the patients exhibited gradual patterns of spatial ITH. In the other half of the patients, on the other hand, the tumors segregated into layers of clones that the authors call tectonic plates which are characterised by different transcriptomic and mutation profiles. They show that in 60% of the patients, different tectonic plates with distinct transcriptional profiles are present where the one with a more advanced (aggressive) phenotype dominates the tumor population. They use this to say that there is Darwinian selection taking place in these tumors.

This is a study with a potential relevance for cancer genomics community. However there are some methodological concerns that precludes the further. Certain aspects of the clonality analysis is not explained. See below.

Reply: Thank you for the concise summary of our work and we really appreciate the encouraging comments recognizing the novel findings of our study to the cancer genomics community. We have addressed the comments in great detail below.

Q1.1 Purity/ploidy estimates for these tumor are not given. Was there a cutoff? This is very basic prior to any type of clonality analysis.

Reply: We agree with the reviewer that tumor purity/ploidy are important parameters for subsequent genomic analysis and have now provided the tumor purity/ploidy estimates in Supplementary Table 2. HCC is a cancer type with high tumor purity. Across our samples, the mean purity is estimate to be 0.69 which is consistent with several previous studies (e.g. PMID: 35382356).

Revision: We have now added these estimates to Supplementary Table 2.

Q1.2 What kind of CNA was performed? Didn't the authors identify any putative oncogenic CNAs?

Reply: 1) We have performed the copy number analysis (CNA) for all the tumor samples using the computational method implemented in package "Sequenza". 2) Since the number of independent patients in our cohort is not large, in order to identify putative oncogenic CNAs, we have performed the driver CNA analysis using the GISTIC algorithm based on the HCC data (LIHC) from The Cancer Genome Atlas (TCGA, n=375).

Revision: We have now added this result as Supplementary Figure 2d.

Q1.3 Any mutation filtering performed prior to Pyclone analysis? Mutations present in low VAFs across many tumors might be artefactual and affect downstream clonality analysis.

Reply: This is a very good question. In our PyClone analysis, we have taken these steps: 1) take a union of all the SNVs across all sectors for a given patient; 2) extract the number of reference and alternative allele count at the SNV positions for all sectors (based on Samtools mpileup). 3) We required the SNVs to have a frequency >0.05 in at least one sector (to filter away possible false positives); 4) We employed PyClone for the clonal inference based on the variant allele frequency, local copy number (inferred from sequenza) as well as tumor purity/ploidy estimates. Taken together, we did perform filtering of low fidelity variants before the PyClone analysis.

Revision: we have added these information to the updated M&M.

Q1.4 Were indels and SNVs analyzed together? I would say this must be the case since the authors have exome data only. However it is not clear from the text. VAF estimates for indels are not as accurate as for SNVs. Might this affect clustering using Pyclone?

Reply: The reviewer is quite right that indel calling is still a non-trivial task, especially accurately estimate the indel variant frequency can be challenging due to subtleties in short read mapping. We are aware of the inaccuracy in indel variant frequency estimation and had similar concerns as the reviewer, so, in the clonality (e.g. PyClone etc) analysis, we have only used SNVs (indel only contributed 6.56% of the total variability) to have good quality input data for clonality inference.

Revision: we have now added this information to the latest M&M.

Q1.5 Are clone trees based on both CNAs and SNV/Indels? Any filtering performed on clusters after Pyclone analysis? Were there clusters with predominantly indels or with SNVs/indels with low CCFs across many tumors of the patient? These might be artefactual as well.

Reply: Thank you for the question. After estimating the clonal composition of the samples (see Q1.3) based on PyClone, we performed further filtering and analysis for estimating the clone trees. 1) We filtered away the clones with only one mutation; 2) We then extracted the top 10 clones with the highest number of mutations and used CITUP to estimate the clonal trees. (Recommended by CITUP program (PMID: 25568283) as larger number of clones will lead to difficult optimization problems as the number of combinations is computationally intractable).

Revision: we have now added these updates to the updated M&M.

Q1.6 General stats on clusters should be provided as a table (number of mutations, CCF in each sample etc)

Reply: Thank you for the suggestion and we agree with the reviewer that

making the data available to the community is important for the field. In addition to making the raw data available to the field, we have compiled a separate file containing all the important information (Supplementary Table 5). The supplementary table contains the mutation, mutational signature, copy number alterations, gene expression as well as the clonal information.

Revision: we have added the Supplementary Table 5 to the latest version of our manuscript.

Q1.7 The final clone trees in S. Fig. 3 are not annotated with putative oncogenic events. Are there subclonal drivers that characterize the patients with punctuated spatial heterogeneity?

Reply: This is a very good question. We have followed the reviewer's suggestion to: 1) We have extracted from a recent integrative analysis where we have combined more than 1300 HCC genomes and identified a list of 62 driver genes for HCC (PMID: 35832070). We have then annotated the driver genes onto the clonal trees for all the patients. When we focused on the patients with spatially punctuated heterogeneity, we indeed found that mutations in *HNF4A* and *DOCK2* are associated with the "more aggressive" clone (denoted as the selected clone) in SH05. Similarly, *ALB* is associated with the selected clone in SH10. For SH06, we observed much evaluated whole genome copy number in the selected subclone (Fig.4f). Taken together, we indeed observed multiple driver events associated with the subclonal selection.

Revision: we have updated Supplementary Figure 3 to annotate the clonal phylogeny with the driver mutations. In addition, we have also added the driver information to the maintext.

On fixation index, sample trees and clone trees

2.1 The authors draw sample trees and clone trees for each patient (shown in panel-b and panel-k in S. Fig. 3). Sample trees are inferred from the binary mutation matrix in panel-d and show which mutations are shared across the samples. Clone trees are generated by running CITUP on the clones identified by Pyclone and would be the best approximation to the underlying subclonal structure. However all the downstream "genetic" analysis is performed on sample trees. The authors use the sample tree (binary mutation matrix) to calculate the fixation index (FST) which is then used to measure the genetic relatedness. What worries me with this is that sample trees and clone trees are not necessarily equivalent especially if the samples are polyphyletic, which is what it looks like from panel-k in S. Fig. 3. The ideal thing to do would be to compare the phylogenetic relationship between the sectors not the genetic relatedness. How would the results look like if the authors used the absence/presence of the clones instead of the absence/presence of the mutations to calculate FST? Clarify the results, emphasize samples'

geographic locations, not limiting to any special clones' spatial distribution.

Reply: Thank you for pointing out potential gaps in our presentation and thank you for the nice suggestions. In the original analysis of the spatial heterogeneity, we used different approaches to demonstrate the Isolation-By-Distance (IBD) relationship. Firstly, we measured the genetic differentiation between samples by calculating the F_{ST} between sectors. F_{ST} measures genetic differentiation based on mutant allele frequency variance across multiple samples. We found that genetic differentiation is positively correlated with physical distances. Secondly, we used PyClone to perform clonal deconvolution of samples, we found that clones generally distributed in a restricted spatial area and physically closer sectors have similar clonal composition. The reviewer is quite right, clonal trees and sample trees might lead to different representations of the genetic relationship of samples and phylogenetic distance/relationship might be an intuitive alternative approach to measure the genetic differences. In the newer version of the manuscript, we have taken the reviewer's suggestion and added the relationship between phylogenetic distances and physical distances, similar to F_{ST} /clonal distance, we also found a positive correlation between phylogenetic distances and physical distances. In other words, the IBD relationship is quite robust across genetic differentiation (F_{ST}), clonal composition (clonal distance) as well as phylogenetic distance when we dissected the spatial heterogeneity using different metrics.

Revision: In light of the reviewer's questions/suggestions. We have now added: 1) new results on the regression between clonal distance and physical distance, as well as the regression between phylogenetic distance and physical distance (Supplementary Figure 4). 2) We have summarized the findings and added a supplementary note 1 "Isolation by distance relationship is robust to the distance metrics" and we have added the new results to the latest manuscript.

Q2.2 If F_{ST} was calculated from the clonal trees then we would know if there were any specific mutations that might relate with the difference in transcriptional profile. Could the authors annotate the clone trees with clonal/subclonal putative oncogenic changes.

Reply and Revision: This is a good suggestion. We used the unbiased estimator of F_{ST} (Fixation index) from Weir and Cockerham 1984 to measure population differentiation among different samples. In this study, we used F_{ST} as genetic distance metric for pairwise samples (sectors). F_{ST} is a measure of genetic differentiation based on allele frequency variance across multiple populations (two or pairwise samples in our study). Here, we treated each sample (sector) as a population of cells, and the proportion of mutant reads at each SNV position is an estimate of mutant allele frequency. To be more precise, for a given SNV position m ($m = 1, 2, \dots, M$) in sample i ($i = 1, 2, \dots, s$) (s is the total number of sectors), we denoted the number of mutant reads and total reads as n_i and N_i respectively. The mutant allele frequency for the

sample i is thus $\tilde{p}_i = \frac{n_i}{N_i}$. The mean number (coverage) and frequency of mutant allele across two samples (e.g. $i, j \in 1, 2, \dots, s$) can be expressed as $\bar{n} = \frac{n_i+n_j}{2}$ and $\bar{p} = \frac{n_i+n_j}{N_i+N_j}$. We apply Weir and Cockerham's estimator of F_{ST} , which accounts for sample size differences. For each position m , the genetic differentiation between two samples ($i, j = 1, 2, \dots, s$) can be calculated as:

$$\widehat{\theta}_m(i, j) = \frac{S^2 - \frac{1}{\bar{n}-1} \left[\bar{p}(1-\bar{p}) - \frac{1}{2} S^2 \right]}{\left[\frac{n_c-1}{\bar{n}-1} \right] \bar{p}(1-\bar{p}) - \left[1 + \frac{(\bar{n}-n_c)}{\bar{n}-1} \right] \frac{S^2}{2}}$$

where, $n_c = 2\bar{n} - \frac{n_i^2+n_j^2}{2\bar{n}}$, $S^2 = \frac{n_i(\tilde{p}_i-\bar{p})^2+n_j(\tilde{p}_j-\bar{p})^2}{\bar{n}}$. When there are M mutations, we estimate F_{ST} across all mutations between pairwise samples as:

$$F_{ST}(i, j) = \frac{\sum_{m=1}^M \widehat{\theta}_m(i, j)}{M}.$$

The reviewer made a very nice suggestion to annotate driver mutations, we have extracted from a recent integrative analysis where we have combined more than 1300 HCC genomes and identified a list of 62 driver genes for HCC (PMID: 35832070). We have now annotated the clonal trees with clonal/subclonal putative oncogenic drivers. We found that mutations in *HNF4A* and *DOCK2* are associated with the "more aggressive" clone (denoted as the "selected clone") in SH05. Similarly, *ALB* is associated with the selected clone in SH10. For SH06, we observed much elevated genome copy number in the selected subclone (Fig.4f). In summary, we indeed observed multiple oncogenic drivers associated with the punctuated spatial heterogeneity. We have now added these results to the latest version of the manuscript.

Q2.3 The authors partition the tumor along different central axes and calculate the FST-ratio, mean FST within a partition divided by mean FST across partitions. They designate patients whose tumors show fluctuations across different partitions as tectonic. I wonder if this analysis is too coarse and if the authors need a finer grid search. Some of the "non-tectonic" tumors do seem to have samples that share a substantial number of mutations (mutation heatmap in S. Fig. 3) and are indeed within physical proximity of each other (S. Fig. 1). For example

- SH03: (T01, T02, T03 T06), (T05, T07, T08, T11, T12), (T14, T15, T19, T20, T21, T22, T25, T26, T27, T29) and (T16, T17, T18, T23, T24)
- SH07: (T05, T06, T11, T12) and (T08, T14, T15, T18, T19)
- SH08: (T02, T03, T07, T08, T10) and (T04, T05, T09, T12, T13)

Reply: Thank you for the nice comment. In the previous version of the manuscript, we discovered that the linear relationship in the FST vs physical

distance regression (e.g. Figure 2a for SH05) had two parallel linear lines, which was due to two separate groups of sectors where sectors from different groups (i.e. between) had much higher genetic differentiation (F_{ST}) than sectors from within groups (i.e. within). The two groups partitioned the tumor from the middle into two halves. We thus hypothesized that tumors with tectonic plates will tend to have high divergence between two halves of the tumors. We thus developed a strategy to partition the tumor along different central axes and calculated the $F_{ST_{ratio}}$ (between vs within divergence, defined as the $F_{ST_{between}}/F_{ST_{within}}$). We found that tumors with tectonic plates tend to have very high $F_{ST_{ratio}}$ similar to SH05. As pointed out by the reviewer, this might be a good exploratory approach, but can be bit coarse.

In the latest version of the manuscript, we have updated the procedure dissecting the spatial heterogeneity. In particular, we have now 1) calculated the phylogenetic distance between tumor sectors and retrieved a distance matrix for each patient. 2) We then used a hierarchical clustering method based on the phylogenetic distance matrix and inferred the clusters for a given number of clusters k ($k=2-5$). 3) In order to have a metric that can be compared across patients, we used F_{ST} to calculate the genetic divergence between and within the clusters. In particular, we employed the Calinski-Harabasz Index (CH-index) which is calculated as $F_{ST_{between}}/F_{ST_{within}}$ to pick the optimal number of clusters for each patient. The reviewer is quite right, we found that clusters generally reside in spatially continuous regions, matching the IBD relationship found earlier (Supplementary Figure 3 and Supplementary Figure 9). Interestingly, we found that a subset of patients (5 out of 10) have much larger genetic differentiation between clusters than within cluster (i.e. one magnitude higher with CH-index > 10) and these five patients are matching those patients with tectonic plates in our original analysis. Thus, we stated that the IBD relationship leads to (genotypically/phenotypically) similar sectors in regional areas (i.e. spatial blocks) and we designated those five patients with very high CH-index as patients with spatially punctuated heterogeneity (SPH).

Revision: We have employed the new procedure presented above and have now rewritten the section on “Spatially punctuated heterogeneity with variegated blocks in HCC” as well as associated M&M.

Other points

Q3.1 Please provide the citation for “genome instability index”.

Reply and revision: We have added the reference for the genome instability index (PMID: 17925008) to the latest version (M&M) of the manuscript.

3.2 On lines 333-335, the authors say all patients with Warring States distribution of clones in the transcriptomic data also have evidence of positive selection as assessed by MOBSTER. However, there are selected clones in non-Warring States patients as well (Fig. 5d). How do the authors explain this?

Reply: Thank you for the question. The reviewer is quite right that we first applied MOBSTER to the subset of patients in the Warring States, subsequently when we extended the analysis to the remaining patients (in non-Warring States), we also discovered evidence of positive selection. We think that the evidence of natural selection is much more pervasive than previously thought. We think there are several factors that can contribute to this observation: 1) As natural selection might operate in the form of “local adaptation” and ongoing selection might be missed if we only survey a fraction of the tumor. We think that the macroscopic approach employed in our study allowed us to “observe” a more extensive landscape of intra-tumor heterogeneity and detect more Darwinian selection in tumors. 2) We think the rapid tumor microenvironmental changes (e.g. different transcriptomic subtypes) might have triggered a strong selection during the history of tumor development. 3) Given the large effective population size of the tumor populations, we think natural selection can operate more actively than what we thought previously.

Revision: we have added these points to the latest version of the discussion. Detecting Darwinian evolution in cancer genomics has mainly focused on detecting convergent evolution (i.e. repeatedly occurring mutations across many patients). Detecting ongoing Darwinian evolution, especially with spatially sampled sectors, are still in its infancy. As these observations on macroscopic heterogeneity are fairly new and this is a nascent area in the field, we have also written a short supplementary note 4 “statistical properties in detecting natural selection in tumor evolution” further discussing these issues.

Q3.3 Throughout the text, when referring to figures please refer to which panel. For example in lines 138 and 141, it should be "Supplementary Fig. 2a" and "Supplementary Fig. 2b", respectively.

Reply and revision: Thank you! We have corrected the citation as suggested.

Q3.4 Please label panel-i S. Fig. 3 with tumor ids.

Reply and revision: Thank you! We have now added the tumor ids to the Supplementary Figure S3.

We have tried our best to address all the questions. These revisions have significantly improved the presentation of our work. We want to thank the reviewer for the constructive comments and we hope you find the revisions to your satisfaction.

Reviewer #3 (Remarks to the Author)

The manuscript from Liu and colleagues collects a mean of 17 samples from each of 13 hepatocellular tumours, and performs exome and RNAseq on each sample. Their analysis is based around a comparison of clonal structure (measured from DNA) and the association with gene expression. They report that tumours often segregate into a small number of large subclones, with distinct gene expression patterns, and further that one of the subclones has a putatively more aggressive gene expression pattern. The conclusion broadly is that measuring spatial heterogeneity gives a new window in cancer evolutionary dynamics, and in liver cancer in particular, that subclonal selection is common.

Spatially resolved tumour heterogeneity is interesting (and a hot topic), and the large dataset of spatially resolved DNA and RNA data is valuable. The authors push a fairly novel line of analysis (though others have looked at DNA-RNA relationships fairly extensively in cancer before), but there are many assumptions and uncertainties in the analysis that leave us feeling unsure of the strength of the conclusions.

Reply: Thank you for the concise summary of our work. We appreciate the encouraging comments and have carefully addressed the questions in great detail in the following section.

Detailed comments:

Q1 The F_{ST} statistic is used to measure genetic distance between populations. It wasn't clear to us how the statistic was calculated (it is not specified in the methods: some precise mathematical formulas should be given). This makes it very hard to assess the meaning of the results.

Reply: We apologize for not including the details of the formula. We used the unbiased estimator of F_{ST} (Fixation index) from Weir and Cockerham 1984 to measure population differentiation among different samples. In this study, we used F_{ST} as genetic distance metric for pairwise samples (sectors). F_{ST} is a measure of genetic differentiation based on allele frequency variance across multiple populations (two or pairwise samples in our study). Here, we treated each sample (sector) as a population of cells, and the proportion of mutant reads at each SNV position is an estimate of mutant allele frequency. To be more precise, for a given SNV position m ($m = 1, 2, \dots, M$) in sample i ($i = 1, 2, \dots, s$) (s is the total number of sectors), we denoted the number of mutant reads and total reads as n_i and N_i respectively. The mutant allele frequency for the sample i is thus $\tilde{p}_i = \frac{n_i}{N_i}$. The mean number (coverage) and frequency of mutant allele across two samples (e.g. $i, j \in 1, 2, \dots, s$) can be expressed as $\bar{n} = \frac{n_i + n_j}{2}$ and $\bar{p} = \frac{n_i + n_j}{N_i + N_j}$. We apply Weir and Cockerham's estimator of F_{ST} , which accounts for sample size differences. For each position m , the genetic differentiation between two samples ($i, j = 1, 2, \dots, s$) can be calculated as:

$$\widehat{\theta}_m(i, j) = \frac{S^2 - \frac{1}{\bar{n} - 1} \left[\bar{p}(1 - \bar{p}) - \frac{1}{2} S^2 \right]}{\left[\frac{n_c - 1}{\bar{n} - 1} \right] \bar{p}(1 - \bar{p}) - \left[1 + \frac{(\bar{n} - n_c)}{\bar{n} - 1} \right] \frac{S^2}{2}}$$

where, $n_c = 2\bar{n} - \frac{n_i^2 + n_j^2}{2\bar{n}}$, $S^2 = \frac{n_i(\bar{p}_i - \bar{p})^2 + n_j(\bar{p}_j - \bar{p})^2}{\bar{n}}$. When there are M mutations, we estimate F_{ST} across all mutations between pairwise samples as:

$$F_{ST}(i, j) = \frac{\sum_{m=1}^M \widehat{\theta}_m(i, j)}{M}$$

Revision: We have added the FST formulation to the latest version of the Materials and Methods.

Q2: The FST statistic requires an assessment of within-sample diversity, which is a problematic thing to measure. The methods seem to imply pyclone was used for this, but do not specify how within and between sample diversity was calculated (SNV distance between deconvolved subclones?). Further, the authors inconsistently switch between pyclone and then MOBSTER methods to measure clone structure – why? We note that MOBSTER corrects for neutral tails: we’re not clear how these tails influence FST calculations (failing to remove them certainly risks introducing spurious clones into the FST analysis).

Reply: Thank you for pointing out the gap in the presentation. We have used allele frequencies from different samples (sectors) as the input for calculating the FST values (Q1 above), which measure differences in allele (SNV) frequencies across samples. Based on the positive correlation between FST and physical distances, we concluded that there is a general trend of isolation-by-distance (IBD) pattern where physically closer sectors are genetically more similar. We have not used PyClone output in the FST calculation.

In the second part, the reviewer pointed out that we used both PyClone and MOBSTER for clonal deconvolution in two different settings. We used PyClone to explore the spatial differences in the clonal structure across multiple sectors. For MOBSTER, we used its function to detect selected subclones (after correcting for neutral tails in a single sector). In other words, we used PyClone to generically deconvolute the clonal structure across multiple samples and MOBSTER was used to specifically detect selected subclones in a given sample.

Revision: We have added the details about the FST calculation in the latest version of the manuscript (M&M and results). We have also clarified the utilities of the PyClone and MOBSTER inference in the maintext.

Q3. We think a simpler way to measure genetic distance would be to use distances measured on phylogenetic trees. However, if phylogenetic

distances are used (which we strongly recommend), then the authors need to redo their tree building, as currently they use neighbour joining which is very crude (essentially clustering) and so is not really a proper phylogenetic method. Maximum parsimony methods seem to be the accepted minimum standard in the cancer field and we suggest the authors use this (or a more sophisticated ML/Bayesian method).

Reply: Thank you for the nice recommendation. We agree with the reviewer that neighbor-joining is only one of the approaches constructing the phylogenetic relationships among the samples. Following the reviewer's recommendation, we have inferred the phylogenetic relationship based on maximum parsimony and we have updated the results in the maintext. We found that the phylogenetic trees constructed based on the distance method (i.e. neighbor-joining principle) are highly similar to the trees inferred using maximum parsimony (Supplementary Figure 8). In addition, as recommended by the reviewer, when we substitute FST (genetic distance) with the phylogenetic distance, we also observed a consistent IBD relationship.

Revision: we have updated the phylogenetic tree to maximum parsimony tree in the maintext as well as the associated M&M. In addition, we have added a new supplementary note 1 ("isolation by distance relationship is robust to the distance metrics") and supplementary note 3 ("the phylogenetic relationship is consistent across different methods").

Q4. (Fig2; line 172 onwards) Because of our confusion over the FST statistic, the tectonic plates section of the manuscript was hard to follow (particularly how plates (subpopulations) were defined and what this means in terms of genetic distance). It seems the authors always divide their samples by slicing in a straight line through the centre of the tumour (line 485); why? This whole section was very unclear and difficult for us to assess.

Reply: We apologize for the gap in the presentation. In the previous version of the manuscript, we discovered that the genetic differentiation (i.e. FST) vs physical distance regression (i.e. Figure 2a for SH05) had two parallel linear lines, which was due to two separate groups of sectors where sectors from different groups (i.e. between) had much higher genetic differentiation (FST) than sectors from within groups (i.e. within). Moreover, the two groups of sectors tend to split from the middle part of the tumor. We thus hypothesized tumors with tectonic plates will tend to have high divergence between two parts of the tumors and we experimented "slicing in a straight line through the centre of the tumour (line 485)... etc". As pointed out by the reviewer, this might not be the most intuitive way to group the sectors.

Revision: We have followed the reviewer's suggestion (Q5 below) to remove the section around "slicing the tumor..." and developed a new approach suggested by the reviewer. The new approach has indeed made our presentation much simpler and clearer. We really appreciate the suggestions.

- **Q5. Again, we felt this analysis would be likely be much clearer if subpopulations were defined based on a phylogenetic analysis directly. Because evolution is always branching, there are always “subclones” so any discretisation of samples into groups is essentially arbitrary: cutting a phylogenetic tree seems to us to be a straightforward and intuitive way to make groupings.**

Reply: We want to thank the reviewer for very nice suggestion. We have followed your advice to: 1) we constructed parsimony trees using the parsimony method (Q3 above from the reviewer). 2) Based on the parsimony tree, we calculated the phylogenetic distance between tumor sectors and retrieved a distance matrix for each patient. 3) We then used a hierarchical clustering method base on the phylogenetic distance matrix and inferred the clusters for a given number of clusters k (k=2-5). 4) In order to have a metric that can be compared across patients, we used FST to calculate the genetic divergence between and within the clusters. In particular, we employed the Calinski-Harabasz Index (CH-index) which is calculated as $FST_{between}/FST_{within}$. Interestingly, we found that a subset of patients (5 out of 10) have much larger CH-index (i.e. CH index larger than 10) and these five patients matched those patients with tectonic plates in our original analysis. We have updated the results in the new version and this new approach indeed made our analysis much simpler and cleaner.

Revision: Following the reviewer’s suggestion, we have now 1) removed the original schema where we partitioned the tumor along the central axis and discovered subset of tumors with high variation along different partitions (Q4 above), 2) we have followed the reviewer’s suggestion and have implemented the new approach by using phylogenetic trees to identify the spatial clusters and identify the subset of patients with tectonic plates. We have now rewritten the section (“Spatially punctuated heterogeneity with variegated blocks in HCC”). We want to thank the reviewer for the insightful comments.

Comments on the rest of the manuscript are made with the caveat that we were very unsure of the soundness of the genetic heterogeneity measurements in the first half of the paper:

Q6. Is the apparent correlation between spatial heterogeneity and transcriptomic heterogeneity corrected for tumour cellularity in each sample? (fig 2h)

Reply: This is a very good question. In addressing the reviewer’s question, we have tried two different approaches: 1) We identified a subset of genes that is likely to be derived from tumor cells (i.e. positively correlated with tumor purity across patients). Using this subset of genes, we calculated the expressional differences and correlated them with physical distances, we found that IBD relationship is robust (Supplementary Figure 5). 2) We performed a multivariate regression between $y(\text{transcriptomic distance}) \sim x_1(\text{physical distance}) +$

x2(purity differences) to check whether the coefficient for the physical distance is still significant when we take into consideration purity differences. As we found in Supplementary Figure 6, the linear relationship is robust when we take into consideration of tumor purity.

Revision: We have now added these two new results (Supplementary Figure 5 and 6) into the latest version of the manuscript. We have also added a new supplementary note 2 (“the IBD relationship at the phenotypic level is robust to confounding factors”) integrating these results (Supplementary Figure 5 and 6).

Q7. How precisely is transcriptomic distance calculated? Mathematical formulas should be given.

Reply and revision: We used $1-\rho$ where ρ is the spearman correlation coefficient of the expression profiles between two samples (PMID: 35382356) as the distance metric. We have now added the formula and references to the latest version of the M&M.

Q8. Is the influence of CNAs on gene expression accounted for in expression distance measurements? (there is typically a roughly linear relationship between CNAs and gene expression)

Reply: This is a very good question. We have explored the answer to this question by using genes outside CNA regions. When we correlated the physical distances vs the divergence in gene expression by restricting the genes to the copy neutral regions, we found that linear (i.e isolation-by-distance) relationship is consistent (Supplementary Figure 7).

Revision: we have now added these results to the latest version of the manuscript and the supplementary note 2 (“the IBD relationship at the phenotypic level is robust to confounding factors”).

Q9. How does immune cell composition differ across the tumours – could this be shown directly (analogous to Fig 2e-g)

Reply and revision: Thanks for the suggestion. We have plotted the immune heterogeneity across the tumor as suggested by the reviewer and we have added the new figure as Supplementary Figure 16.

Q10. The logic that putatively aggressive clones are often larger than putatively benign clones seems reasonable evidence of positive selection. Does genetic data back this up – are there additional known drivers, or $dN/dS > 1$ in the positively selected clone? If a phylogenetic analysis is done, are the larger clones always phylogenetically newer, therefore faster growing? (relatedly, the implied association between SNV burden and aggression should be shown.)

Reply: This is a nice suggestion. We have followed the reviewer’s suggestion to: 1) We have extracted from a recent integrative analysis (PMID: 35832070,

from our own group) where we have combined more than 1300 HCC genomes and identified a list of 62 driver genes for HCC. 2) We have then annotated the driver genes onto the clonal trees for all the patients. When we focused on the patients with punctuated spatial heterogeneity, we indeed found that mutations in *HNF4A* and *DOCK2* are associated with the “more aggressive” clone (denoted as the selected clone) in SH05. Similarly, *ALB* is associated with the selected clone in SH10. For SH06, we observed much elevated copy number profile in the selected subclone (Fig.4f). So, we indeed observed multiple driver events associated with the subclonal selection.

In addition, we have now labelled the nonsynonymous (A) and synonymous (S) changes on the clonal trees. We indeed observed a few patients to have more derived clones being the putatively aggressive clones. However, the number of patients is relatively small for now, it hasn't not yet reached statistical significance. A larger set of patients will be need to definitively answer this question and we will pursue this line further in a future study.

Revision: We have now updated the Supplementary Figure 3 by adding the driver events as well as A/S ratio. In addition, we have added the new results on driver events to the maintext.

Q11. The tree balance statistics (Fig 5d) don't seem to show a statistically significant enrichment of signals of selection in “warring states” tumours -> the authors seem to imply this is the case? If they aren't enriched in cases where selection is (most) evident, how can we be sure that the statistics are not overcalling selection?

Reply: this is a very good question. When the subclones expanded in the tumor, the advantageous clones will occupy a larger space and will also lead to a tree with imbalanced tree shape. The reviewer is quite right, we didn't find the tree-imbalance test to be significantly enriched in patients in warring states. There are two major reasons: 1) the tree imbalance test heavily depends on the sample size (i.e. number of sectors taken for each tumor). Right now, the number of sectors we took per tumor varies a lot between patients. Statistical power will also likely contribute to the “no enrichment”. 2) the number of patients is not large enough and statistical enrichment will require larger patient cohort than the current study.

We think there are several factors that can contributed to the pervasive Darwinian selection: 1) As natural selection might operate in the form of “local adaptation” and ongoing selection might be missed if we only survey a fraction of the tumor. We think that the macroscopic approach employed in our study allowed us to “observe” a more extensive landscape of intra-tumor heterogeneity. 2) We think the rapid tumor microenvironmental changes (e.g. different transcriptomic subtypes) might have triggered a strong selection during the history of tumor development. 3) Given the large effective population size of the tumor populations, we think natural selection can operate more

actively than what we thought previously.

Revision: we have added these points to the latest version of the discussion. Detecting Darwinian evolution in cancer genomics has mainly focused on detecting convergent evolution (i.e. repeatedly occurring mutations across many patients). Detecting ongoing Darwinian evolution, especially with spatially sampled sectors, are still in its infancy. As these observations on macroscopic heterogeneity are fairly new and this is a nascent area in the field, we have also written a short supplementary note 4 “statistical properties in detecting natural selection in tumor evolution” further discussing these issues.

Q12: We found the “tectonic plates” terminology unhelpful – couldn’t these be more simply described as “subpopulations” or even “subclones”? Relatedly, tumours are 3D but tectonic plates are essentially a 2D surface, so this feels a misleading analogy. The phrase “punctuated spatial heterogeneity” is also confusing, as the word punctuated implies a temporal aspect, which we don’t think is meant. Similarly the “Warring States” terminology seems unnecessary (though it is a nice metaphor): “competing subpopulations” would be simpler.

Reply and Revision: This is an interesting suggestion. We have taken the advice from the reviewer and have replaced “tectonic plates” with “spatial blocks” or “spatially variegated blocks”. Moreover, we have also replaced “Warring States” with “spatially competing populations”. Regarding “punctuated spatial heterogeneity”, we searched for words that can expressed the meaning of “punctuated” or noncontinuous and didn’t find a good alternative word expressing the meaning. We noticed that punctuation/punctuated also have the meaning of rapid/noncontinuous. We have then used “spatially punctuation heterogeneity” to describe the spatially noncontinuous heterogeneity in our study.

Revision: We have now updated the terminology across the manuscript.

We want to take this opportunity to thank the reviewer for the very nice and constructive comments. We have tried our best to address all the questions. These revisions have significantly improved the presentation of our work. We hope you find the revisions to your satisfaction.

Reviewer #4 (Remarks to the Author): Expert in hepatocellular carcinoma genomics

Liu et al performed genotypic and phenotypic analysis of the spatial heterogeneity of 13 hepatocellular carcinomas in 235 spatially informed sectors in an attempt to obtain spatial distribution patterns. The authors discovered punctuated spatial heterogeneity (PSH) in 50% HCC patients and found a spatial distribution named “Warring States” by a virtual deconvolution analysis of transcriptomic profiles from TCGA LIHC cohort and the present study cohort, challenging the popular theory of neutral evolution. The extensive spatial sampling in this study provides an important resource for studying tumor evolutionary patterns at the macroscopic level and the analysis conducted are sound. However, my major concern is that the current study mainly analyzes some phenomena and lacks in-depth mechanism exploration. In addition, this study does not break through the conventional multiregional sequencing strategy, but studies more sectors of tumor specimens. In fact, spatial transcriptome sequencing can provide more information.

Reply: Thank you for the concise summary of our study. We have addressed individual questions in detail below.

Comments:

Q1. Somatic mutations, copy number variations, gene expression matrix and PyClone results for all the samples should be included as supplementary datasets, which are essential for other researchers to reproduce the authors’ findings. The mutation signature activity and mutation burden should also be listed in a supplementary table.

Reply and revision: Thank you for the suggestions. We agreed with the reviewer that it is valuable to make this dataset available to the community. Taking the reviewer’s advice, we have now compiled the somatic mutations, copy number variations, gene expression matrix, PyClone results in Supplementary Table 5. We have now included Supplementary Table to the latest version of the maintext.

Q2. In lines 485-490, Fixation index (FST) is a well-known method in population genetics research and is based on calculating the variance of alleles in a population. However, the use of FST in this study is simplified and based on somatic variants distance (I guess) or transcriptomic distance. Authors should provide a formula to describe the details of calculating the FST in this study.

Reply: Thank you for pointing out the gap in the presentation. The reviewer is quite right that FST is a classical population genetic metric measuring the population differentiation in allele frequencies. We used the unbiased estimator of FST (Fixation index) from Weir and Cockerham 1984 to measure population differentiation among different samples. In this study, we used F_{ST} as genetic distance metric for pairwise samples (sectors). F_{ST} is a measure of genetic

differentiation based on allele frequency variance across multiple populations (two or pairwise samples in our study). Here, we treated each sample (sector) as a population of cells, and the proportion of mutant reads at each SNV position is an estimate of mutant allele frequency. To be more precise, for a given SNV position m ($m = 1, 2, \dots, M$) in sample i ($i = 1, 2, \dots, s$) (s is the total number of sectors), we denoted the number of mutant reads and total reads as n_i and N_i respectively. The mutant allele frequency for the sample i is thus $\tilde{p}_i = \frac{n_i}{N_i}$. The mean number (coverage) and frequency of mutant allele across

two samples (e.g. $i, j \in 1, 2, \dots, s$) can be expressed as $\bar{n} = \frac{n_i + n_j}{2}$ and $\bar{p} = \frac{n_i + n_j}{N_i + N_j}$. We apply Weir and Cockerham's estimator of F_{ST} , which accounts for sample size differences. For each position m , the genetic differentiation between two samples ($i, j = 1, 2, \dots, s$) can be calculated as:

$$\widehat{\theta}_m(i, j) = \frac{S^2 - \frac{1}{\bar{n} - 1} \left[\bar{p}(1 - \bar{p}) - \frac{1}{2} S^2 \right]}{\left[\frac{n_c - 1}{\bar{n} - 1} \right] \bar{p}(1 - \bar{p}) - \left[1 + \frac{(\bar{n} - n_c)}{\bar{n} - 1} \right] \frac{S^2}{2}}$$

where, $n_c = 2\bar{n} - \frac{n_i^2 + n_j^2}{2\bar{n}}$, $S^2 = \frac{n_i(\tilde{p}_i - \bar{p})^2 + n_j(\tilde{p}_j - \bar{p})^2}{\bar{n}}$. When there are M mutations, we estimate F_{ST} across all mutations between pairwise samples as:

$$F_{ST}(i, j) = \frac{\sum_{m=1}^M \widehat{\theta}_m(i, j)}{M}$$

For the distance based on RNA profiles, we have calculated the distance as $1 - \rho$ where ρ is the correlation coefficient of the expression profiles between two samples.

Revision: we have added the exact formulation to the latest version of the manuscript.

Q3. In lines 499-503, the authors mention how to cluster the transcriptome landscape, but miss the method of classifying tectonic plates based on spatial genetic heterogeneity. As shown in Figure 2, the details of clustering spatial genetic level are more important but unclear in Methods section.

Reply: we apologize for the gap in our presentation. In the previous version of the manuscript, we discovered that the genetic differentiation (i.e. F_{ST}) vs physical distance regression (i.e. Figure 2a for SH05) had two parallel linear lines, which was due to two separate groups of sectors where sectors from different groups (i.e. between) had much higher genetic differentiation (F_{ST}) than sectors from within groups (i.e. within). Moreover, the two groups of sectors tend to split from the middle part of the tumor. We thus hypothesized

tumors with tectonic plates will tend to have high divergence between two parts of the tumors and we experimented a procedure to partition the tumor along the center and identified patients with high differences between two halves of the tumor.... Even though this is a good self-developed exploratory method, it will benefit a lot if we could employ more community developed approaches.

In the new version of the manuscript, we have updated the approach in clustering the tectonic plates. In particular, we have followed these procedures: 1) calculated the phylogenetic distance between tumor sectors and retrieved a distance matrix for each patient. 2) We then used a hierarchical clustering method base on the phylogenetic distance matrix and inferred the clusters for a given number of clusters k ($k=2-5$). 3) In order to have a metric that can be compared across patients and allow us to pick the optimal number of clusters, we used F_{ST} to calculate the genetic divergence between and within the tectonic plates. In particular, we employed the Calinski-Harabasz Index (CH-index) which is calculated as $F_{ST_{between}}/F_{ST_{within}}$ to pick the optimal number of clusters for each patient. We found that clusters generally reside in spatially continuous regions, matching the IBD relationship found earlier (Supplementary Figure 3 and 9). Interestingly, we found that a subset of patients (5 out of 10) have much larger genetic divergence between clusters than within cluster (i.e. one magnitude higher with CH-index > 10) and these five patients are matching those patients with tectonic plates in our original analysis. We thus have updated the results across the manuscript.

Revision: we have now rewritten the results in section “Spatially punctuated heterogeneity with variegated blocks in HCC”. We have also added the details of clustering into tectonic plates into the latest version of the M&M.

Q4. It is confused about how to define the distance relationship (between or within) in Figure 2a. If I understand correctly, the classification of sample pairs (Figure 2a) is based on the clustering of spatial genetic heterogeneity (Figure 2e-g) after selecting patients with IBD (Figure 2d). Please consider reordering Figure 2.

Reply: We apologize for the gap in the presentation. The reviewer is quite right, the “within” and “between” was referring to pairs within/between tectonic plates identified in patient SH05. As pointed out by the reviewer, this result is placed too early in the presentation (it was used before the cluster (i.e. between and within) is actually defined).

Revision: In the latest version of the manuscript, we have removed the within/between label and emphasized the logic when we presented tectonic plates in Figure 2e-g.

Q5. Please add color legend in Figure 3c.

Reply and revision: We have now added the color legend as suggested.

Q6. In line 267-269, the authors conclude that the hybrid subtype has an

immune-exclusive status (Figure 3e and I). Please provide more evidence.

Reply: Thank you for the suggestion. In addition to the tumor micro-environmental estimates using ESTIMATE in the previous version, we have also performed immune deconvolution using the Consensus TME method. We found that the hybrid subtype has lower immune infiltration (Supplementary Figure 11). Together with the lower immune infiltration found in Figure 3e (NMF deconvolution output) and 3i (output from ESTIMATE), we consistently found that the hybrid subtype has an immune exclusion status.

Revision: We have added these new results to the latest version of the manuscript as well as Supplementary Figure 11.

Q7. Please provide the comparison group for the differential expression analysis in Figure 4b, e and I. Proliferation vs Hybrid? Or Hybrid vs Proliferation? What the transcriptomic subtypes are in SH12?

Reply and revision: Thank you for the suggestion. we have updated the figure 4 following the reviewer's suggestion.

We have tried our best to address all the questions. These revisions have significantly improved the presentation of our work. We would like to thank the reviewer for the constructive comments and we hope you find the revisions to your satisfaction.

Reviewers' comments:

Reviewer #2 (Remarks to the Author):

Q1.1 Purity/ploidy estimates for these tumor are not given. Was there a cutoff? This is very basic prior to any type of clonality analysis.

REVIEWER: Satisfactory

Q1.2 What kind of CNA was performed? Didn't the authors identify any putative oncogenic CNAs?

Revision: We have now added this result as Supplementary Figure 2d.

REVIEWER: Satisfactory

Q1.3 Any mutation filtering performed prior to Pyclone analysis? Mutations present in low VAFs across many tumors might be artefactual and affect downstream clonality analysis.

REVIEWER: Satisfactory

Q1.4 Were indels and SNVs analyzed together? I would say this must be the case since the authors have exome data only. However it is not clear from the text. VAF estimates for indels are not as accurate as for SNVs. Might this affect clustering using Pyclone?

REVIEWER: Satisfactory

Q1.5 Are clone trees based on both CNAs and SNV/Indels? Any filtering performed on clusters after Pyclone analysis? Were there clusters with predominantly indels or with SNVs/indels with low CCFs across many tumors of the patient? These might be artefactual as well.

REVIEWER: My question is not addressed here. Are the clone trees based on copy number aberrations? From their explanation above it looks like they are SNV-based. Are the SNV-based phylogenies deduced by Pyclone supported by the subclonal structure when copy number profiles are compared? This would help us understand what CNAs are shared across all samples in a patient (trunk) vs those that are subclonal.

Q1.6 General stats on clusters should be provided as a table (number of mutations, CCF in each sample etc)

REVIEWER: I am afraid it is not clear what is presented in S. Table 5. Ideally one would provide a table should contain the number of SNVs in a cluster and the CCF in each sample together with stdev of the CCF in each sample.

Q1.7 The final clone trees in S. Fig. 3 are not annotated with putative oncogenic events. Are there subclonal drivers that characterize the patients with punctuated spatial heterogeneity?

REVIEWER: Satisfactory but still we don't know how the CNAs are distributed along these trees. See my point Q1.5.

On fixation index, sample trees and clone trees

2.1 The authors draw sample trees and clone trees for each patient (shown in panel-b and panel-k in S. Fig. 3). Sample trees are inferred from the binary mutation matrix in panel-d and show which mutations are shared across the samples. Clone trees are generated by running CITUP on the clones identified by Pyclone and would be the best approximation to the underlying subclonal structure. However all the downstream "genetic" analysis is performed on sample trees. The authors use the sample tree (binary mutation matrix) to calculate the fixation index (FST) which is then used to measure the genetic relatedness. What worries me with this is that sample trees and clone trees are not necessarily equivalent especially if the samples are polyphyletic, which is what it looks like from panel-k in S. Fig. 3. The ideal thing to do would be to compare the phylogenetic relationship between the sectors not the genetic relatedness. How would the results look like if the authors used the absence/presence of the clones instead of the absence/presence of the mutations to calculate FST? Clarify the results, emphasize samples' geographic locations, not limiting to any special clones' spatial distribution.

REVIEWER: Satisfactory

Q2.2 If FST was calculated from the clonal trees then we would know if there were any specific mutations that might relate with the difference in transcriptional profile. Could the authors annotate the clone trees with clonal/subclonal putative oncogenic changes.

REVIEWER: Satisfactory

Q2.3 The authors partition the tumor along different central axes and calculate the FST-ratio, mean FST within a partition divided by mean FST across partitions. They designate patients whose tumors show fluctuations across different partitions as tectonic. I wonder if this analysis is too coarse and if the authors need a finer grid search. Some of the "non-tectonic" tumors do seem to have samples that share a substantial number of mutations (mutation heatmap in S. Fig. 3) and are indeed within physical proximity of each other (S. Fig. 1). For example

- SH03: (T01, T02, T03 T06), (T05, T07, T08, T11, T12), (T14, T15, T19, T20, T21, T22, T25, T26, T27, T29) and (T16, T17, T18, T23, T24)

- SH07: (T05, T06, T11, T12) and (T08, T14, T15, T18, T19)
- SH08: (T02, T03, T07, T08, T10) and (T04, T05, T09, T12, T13)

REVIEWER: Satisfactory

Other points

Q3.1 Please provide the citation for “genome instability index”.

REVIEWER: Satisfactory

3.2 On lines 333-335, the authors say all patients with Warring States distribution of clones in the transcriptomic data also have evidence of positive selection as assessed by MOBSTER. However, there are selected clones in non-Warring States patients as well (Fig. 5d). How do the authors explain this?

REVIEWER: Satisfactory

Q3.3 Throughout the text, when referring to figures please refer to which panel. For example in lines 138 and 141, it should be "Supplementary Fig. 2a" and "Supplementary Fig. 2b", respectively.

REVIEWER: Satisfactory

Q3.4 Please label panel-i S. Fig. 3 with tumor ids.

Reply and revision: Thank you! We have now added the tumor ids to the Supplementary Figure S3.

REVIEWER: Satisfactory

Reviewer #3 (Remarks to the Author):

We thank the authors for their responses. We continue to view the question addressed (genotype-phenotype relations) as interesting and timely, and see the value in the datasets that the authors have generated, but sadly also continue to have major reservations about the analysis and so we do not think that the study provides good evidence for the conclusions reached.

1. We're grateful for the specification of the FST statistic. Unfortunately, we are now even more skeptical that the FST represents genetic diversity in cancer in a meaningful way. Essentially FST was designed to look at allele segregation in sexually evolving populations, and uses allele frequency in the population.

We do not think it is reasonable to directly translate this idea to look at asexual evolution within a single sample. In detail:

Cockerham Weir's F_{ST} was designed for the Wright-Fisher ideal population. It assumes that alleles (p and q summing to 1 at Hardy-Weinberg equilibrium) are present in an ancestral population which subsequently may be passed on by random mating (panmixia), or non-random mating (forming distinct sub-population structure). The statistic is based on the comparison of expected heterozygosity (expressed as p times q or $p(1-p)$) within and between populations. The authors here define p as the number of mutant reads / total reads, (VAF). The F_{ST} equation presented is therefore the $VAF \times (1-VAF)$ compared to the average $VAF \times (1-VAF)$ across two samples. This formulation makes no phenomological sense to us: as the quantity $1-VAF$ compared to VAF seems non-sensical as a key indicator of diversity. Further, the final part of the F_{ST} calculation involves a summation of the above across all possible loci. In a Wright Fisher population where loci are separated during gamete formation, this provides an independent estimate across multiple markers. In cancer lineages, mutations are expected to be inherited together or sequentially. Therefore, the null expectation would be an increasing F_{ST} over time and space as mutations accrue higher VAFs in different cancer lineages. Whereas in cancer, the absence of recombination guarantees some sub-population structure if applying the F_{ST} statistic. (we note that the other parts of the F_{ST} equation are concerned with unequal population sizes (or in this case differing number of reads) – and this feels uncomfortable as reads are not the same as individuals).

If the utility of F_{ST} used as the authors have done can be proven, then we think it is also essential to further prove that it is robust to considerations such as cancer cell content (purity), ploidy, ploidy calling errors, and subclonal structure. We suggest computational simulations are the only reasonable way to do this.

2. Related to the previous, we feel there is an inherent conflict between the use of the F_{ST} statistic (which ignores subclonal structure), and then later pyclone and MOBSTER (which both explicitly detect subclonal structures). Are these two broadly distinct approaches giving consistent results?

Further, the authors state that only use MOBSTER for detecting selection, but do not address the fact that MOBSTER and pyclone handle neutral tail mutations differently, potentially biasing the results. This additional major discrepancy in the methods needs to be fully addressed.

3. We appreciate the new use of maximum parsimony to construct phylogenetic trees, and also the revised classification of tumour regions. We don't follow though why the clustering of phylogenetic distances is needed – this seems to throw away phylogenetic information as two sets of samples arranged in distinct clades could have similar phylogenetic distances within each clade, so would be clustered together. Why not just cut the tree into clades directly?

4. Further, given the authors are now using a phylogenetic analysis, why do they need the F_{ST} statistic at all? Why not just use phylogenetic distances that they have already calculated?

5. A major conclusion of the manuscript is that clonal selection is pervasive in a subset of these cancers. The data don't seem to support this as Fig 5d shows no enrichment of the selection signals between tumours with or without the "spatially competing distribution" of subclones. How information is combined across the different statistics is not explained, nor how the large number of discrepancies between statistics should be interpreted. We feel some level of uncertainty in the conclusions should be robustly acknowledged at minimum.

Reviewer #4 (Remarks to the Author):

I am satisfied with the revised manuscript.

Reviewer #2 (Remarks to the Author):

Q1: Are the clone trees based on copy number aberrations? From their explanation above it looks like they are SNV-based. Are the SNV-based phylogenies deduced by PyClone supported by the subclonal structure when copy number profiles are compared? This would help us understand what CNAs are shared across all samples in a patient (trunk) vs those that are subclonal.

Reply: the reviewer is quite right that the clone trees are based on SNVs. We first used PyClone to compute the cancer cell fraction (CCF) across the sectors and subsequently used the CCF distribution to cluster the SNVs into clones before resolving their clonal structure. When we computed the CCF of the SNVs in each sector, sector specific CNV profiles were used for computing the CCF. So, variation in CNVs profiles across sectors was taken into account in the PyClone inference. In HCC, most of the CNVs, especially the large chromosomal CNVs, tend to be early events and are shared between sectors (Supplementary Figure 2d).

Q2. General stats on clusters should be provided as a table (number of mutations, CCF in each sample etc). I am afraid it is not clear what is presented in S. Table 5. Ideally one would provide a table should contain the number of SNVs in a cluster and the CCF in each sample together with stdev of the CCF in each sample.

Reply and revision: Thank you for the suggestion. We have provided the raw data from the current analysis for the research community s.t. our data can be further utilized. The raw data include the list of mutations, mutational signatures, CNV profiles, RNAseq (raw counts) as well as the clonal structure (pls see Supplementary Table S5).

Reviewer #3 (Remarks to the Author):

We thank the authors for their responses. We continue to view the question addressed (genotype-phenotype relations) as interesting and timely, and see the value in the datasets that the authors have generated, but sadly also continue to have major reservations about the analysis and so we do not think that the study provides good evidence for the conclusions reached.

Reply: We want to thank the reviewer for recognizing the importance of studying genotypic and phenotypic spatial heterogeneity. In this work, we aimed to dissect the macroscopic intra-tumor heterogeneity (ITH), trying to get a global overview of the ITH across the tumor. Even though ITH has been extensively studied, the macroscopic spatial ITH is not yet explored. We appreciate the constructive comments you have given us in the first review and we found that the revision had significantly improved our presentation. We realized that we are at the interface between evolutionary genetics and cancer genomics. At this interdisciplinary area, both approaches from molecular evolution (e.g. phylogenetic trees) and population genetics (e.g. FST) were often applied, which might lead to some discrepancies in understanding the results. Moreover, by reading the further comments from the reviewer, we realized that we might not have fully understand the reviewer's suggestions, which have led to the gap in the presentation. We apologize for those "misunderstandings". In this revision, we have performed further analysis to address the questions from the reviewer. We think our study not only generated one of the largest datasets for studying macroscopic spatial ITH, but also revealed quite novel observations about spatial ITH. We hope you like our latest revision.

Q1. We're grateful for the specification of the FST statistic. Unfortunately, we are now even more skeptical that the FST represents genetic diversity in cancer in a meaningful way. Essentially FST was designed to look at allele segregation in sexually evolving populations, and uses allele frequency in the population. We do not think it is reasonable to directly translate this idea to look at asexual evolution within a single sample. In detail:

Cockerham Weir's FST was designed for the Wright-Fisher ideal population. It assumes that alleles (p and q summing to 1 at Hardy-Weinberg equilibrium) are present in an ancestral population which subsequently may be passed on by random mating (panmixia), or non-random mating (forming distinct sub-population structure). The statistic is based on the comparison of expected heterozygosity (expressed as p times q or $p(1-p)$) within and between populations. The authors here define p as the number of mutant reads / total reads, (VAF). The FST equation presented is therefore the $VAF \times (1-VAF)$ compared to the average $VAF \times (1-VAF)$ across two samples. This formulation makes no phenomological sense to us: as the quantity $1-VAF$ compared to VAF seems non-sensical as a key indicator of diversity. Further, the final part of the FST calculation involves a summation of the above across all possible loci. In a Wright Fisher population where loci are separated during gamete formation, this provides an independent estimate across multiple markers. In cancer lineages, mutations are expected to be inherited together or sequentially. Therefore, the null expectation would be an increasing FST over time

and space as mutations accrue higher VAFs in different cancer lineages. Whereas in cancer, the absence of recombination guarantees some sub-population structure if applying the FST statistic. (we note that the other parts of the FST equation are concerned with unequal population sizes (or in this case differing number of reads) – and this feels uncomfortable as reads are not the same as individuals).

If the utility of FST used as the authors have done can be proven, then we think it is also essential to further prove that it is robust to considerations such as cancer cell content (purity), ploidy, ploidy calling errors, and subclonal structure. We suggest computational simulations are the only reasonable way to do this.

Reply: Thank you for the careful thoughts and we appreciate the comments given by the reviewer. As a colleague from evolutionary biology, we agree with most of the comments from the reviewer.

- a) We agree with the reviewer that the confounding factors such as purity/ploidy need to be properly take into account. In light of the reviewer's comments, we have conducted a literature review and employed a computational procedure from a recent study taking into account purity/ploidy when computing the FST value¹.
- b) The reviewer is quite right that the original theory for FST is developed based on the Wright-Fisher model and the canonical formulation was often based on sexually reproducing systems (i.e. organismal evolution). Later on, the utility of FST has been extended to many other settings (e.g. non-recombining mtDNA and many other systems including cancer¹).

One of the further extension of FST is that it is a metric that measures the proportion of total variation (i.e. allele frequency differences) that is explained by the between sample variance (i.e. $\text{Var}_{\text{between}}/\text{Var}_{\text{total}}$). In this study, we have used FST as a distance metric to measure the level of differentiation in allele frequency between samples. Across the manuscript, we have employed FST in two places:

- a) Isolation-By-Distance (IBD) pattern of spatial ITH: We found that there is a linear relationship between physical distance and genetic differentiation (i.e. FST) between samples. In addition to FST, we have also evaluated several other distance metrics such as hamming distance and clonal compositional distance, we confirmed that such linear relationship is consistent across multiple distance metrics.
- b) Spatially punctuated heterogeneity: after clustering the samples into clusters (spatial blocks), we have used FST to calculate the genetic divergence between and within spatial blocks. Subsequently, we picked the optimal number of clusters based the ratio of between/within block variation (i.e. CH index). We have previously chosen to employ FST as the distance measure mainly because it is a metric that is comparable across patients.

Revision: In light of the reviewer's comments, we have updated FST formulation in the first place (a) by taking into account confounding factors such as purity/ploidy. For the second place (b), we have taken away the utility of FST, but substitute with a similar procedure suggested by the reviewer (see Q3-4 below). We have chosen to maintain FST because the canonical formulation of IBD in Evolutionary Genetics has been defined based on FST and physical distance. We hope the reviewer can agree with the new layout from

the current revision (please see Q3-4 for further details).

Q2. Related to the previous, we feel there is an inherent conflict between the use of the FST statistic (which ignores subclonal structure), and then later pyclone and MOBSTER (which both explicitly detect subclonal structures). Are these two broadly distinct approaches giving consistent results?

Further, the authors state that only use MOBSTER for detecting selection, but do not address the fact that MOBSTER and pyclone handle neutral tail mutations differently, potentially biasing the results. This additional major discrepancy in the methods needs to be fully addressed.

Reply: We appreciate the careful thoughts from the reviewer. The FST statistic is a metric employed in Population Genetics for measuring the level of allelic frequency differences across samples. The clonal deconvolutional approaches clustered the mutations into groups (i.e. clones) and are often utilized in the cancer genomics field. Both approaches are “summaries” of the allele frequencies across samples. In our study, we have employed both approaches to study the relationship between genotypic difference and physical distance. For example, when we calculate the genetic divergence (i.e. FST) as well as compositional distance at the clonal level between sectors, we found that multiple distance metrics are highly correlated and have a positive correlation with physical distance (i.e. IBD relationship). Thus, the two approaches are giving concordant results.

Regarding the utility of MOBSTER and PyClone, both of these methods are utilized to cluster the SNVs into groups. PyClone is a generic clonal deconvolution method. It partitions mutations of similar frequencies and is more sensitive to detect subclones (neutral or selected). It also has the advantage of jointly modeling allele frequencies across multiple samples. PyClone serves as the tool of choice for dissecting ITH in our study. MOBSTER also deconvolutes the mutations based on the spectrum (distribution) of mutations, however it constructed a mixture model to control for neutrally evolving cell populations. It detects shifted spectrum (i.e. a shifted peak of mutations) while assigning most of the low frequency variants to neutral tails. Since MOBSTER emphasized more on the selected clones, it is conservative on the number of detected subclones (e.g. Figure 1).

Figure 1: Comparison of clonal structure between PyClone and MOBSTER.

Alluvial plot of the clustering of mutations between PyClone and MOBSTER. Each line is a mutation. The boxes on both ends mark the identified clones. For example, MOBSTER identifies both neutral tails (bottom box) and a clonal peak (upper box, C1). PyClone, on the other hand, identifies multiple subclones (many boxes, on the right). The clonal cluster (C1) identified by MOBSTER corresponds well to a major clone identified by PyClone. Because PyClone models allele frequencies across multiple samples, it can further partition the neutral tails from a single sample into multiple subclones.

Revision: We have added the clonal comparison as the Supplementary Figure 13.

Q3. We appreciate the new use of maximum parsimony to construct phylogenetic trees, and also the revised classification of tumour regions. We don't follow though why the clustering of phylogenetic distances is needed – this seems to throw away phylogenetic information as two sets of samples arranged in distinct clades could have similar phylogenetic distances within each clade, so would be clustered together. Why not just cut the tree into clades directly?

Q4. Further, given the authors are now using a phylogenetic analysis, why do they need the FST statistic at all? Why not just use phylogenetic distances that they have already calculated?

Reply: Thank you for the constructive comments. We realized that we might not have fully understand the reviewer's comments earlier (we apologize for this). We would like to address these two questions together as they are highly correlated. In displaying the spatial heterogeneity, we have used multiple approaches including FST, phylogeny as well as the PCA map. We found that the spatial clustering pattern are often highly similar across multiple approaches. In order to select the optimal number of clusters, we have previously

chosen to use FST because FST is a metric that is more comparable across patients, while the phylogenetic/hamming distances can be strongly influenced by the number mutations across patients. In light of the reviewer's question (i.e. Q1 above), we have revised the manuscript as:

Revision: We realized that the reviewer was suggesting us to use the phylogenetic distance and cut the tree into optimal clusters. We agree that this can be a good alternative approach. In this revision, we have taken the reviewer's suggestion and have updated the results based on the phylogenetic tree. In the latest revision, we have cut the phylogeny (i.e. Maximum Parsimony tree suggested by the reviewer in the previous revision) and picked the optimal number of clusters based on the CHindex ($\text{Var}_{\text{between}}/\text{Var}_{\text{within}}$). We have systematically updated our results and associated Materials and Methods.

5. A major conclusion of the manuscript is that clonal selection is pervasive in a subset of these cancers. The data don't seem to support this as Fig 5d shows no enrichment of the selection signals between tumours with or without the "spatially competing distribution" of subclones. How information is combined across the different statistics is not explained, nor how the large number of discrepancies between statistics should be interpreted. We feel some level of uncertainty in the conclusions should be robustly acknowledged at minimum.

Reply: Thank you for the critical comments. We would like to explain the context of the selection test by "replaying the logic of the manuscript:

- 1) By extensive spatial sampling across the tumors, we first discovered an Isolation-By-Distance (IBD) pattern where physically closer sectors are genotypically and phenotypically more similar. More interestingly, when we clustered the genotypic/phenotypic profile of the tumor sectors, we found that tumors have spatially punctuated heterogeneity (SPH) where tumor sectors segregate into spatially variegated blocks with large genotypic and phenotypic divergence.
- 2) This SPH together with variegated blocks raised an interesting question: what could have led to this spatial pattern. One of the possible reasons for the variegated blocks is natural selection where local adaptation drive genetic differentiation between populations. When applying the neutrality tests (e.g. MOBSTER or tree imbalance test) to the cohort, we found that the majority, if not all, of the tumors have strong evidence of natural selection.
- 3) In order to further dissect the spatial phenotypic pattern, we applied the deconvolution method to the transcriptomic landscape and discovered that a significant subset of patients (n=4 out of 13) have multiple transcriptomic subtypes occupying different spatial blocks within a single tumor (termed as Spatially Competing Distribution, SCD). We employed a simple test (termed as the proportion test) where we compared the sizes of geographic areas taking different transcriptomic subtypes. We found that the derived transcriptomic subtype (i.e. more aggressive) which appear later in time, always occupies a larger geographic area. These observations further confirmed the existence of natural selection in these patients.

The intuition for our observation (pervasive natural selection) is that, when the population size of the tumor is really large, natural selection can be extremely powerful (i.e. even small selective coefficient s be translated into very large value of $S=4Ns$). We might not detect natural selection when we sample very limited number of sectors (e.g. one sector per tumor in the TCGA), but when we sample extensively across the tumor like what we did in our study, the chance of detecting natural selection will be a lot higher.

The reviewer raised an interesting question that, we didn't observe "stronger evidence" of natural selection in patients with the SCD distribution. There are several important reasons for this: 1) SCD patients, are defined by the changes at the phenotypic level. However, the existing methods for detecting natural selection are mainly based on the genetic changes. We still lack proper methods modeling and detect natural selection at the phenotypic level. When we used the Sackin S statistic to test the imbalance of the transcriptomic (RNA) tree, we did rejected neutrality for all SCD patients. 2) The number of patients in SCD and non-SCD is not large, even though we do detect important genetic changes associated with the genetic divergence (for example, we observed much elevated genome copy number in the selected subclone in SH06 in Fig.4f). These changes are quite "individualized" and is not statistically significant. 3) Thirdly, we think patients with SCD distribution are exceptional cases of tumor evolution where there are often two dominant sub-populations with large genotypic and phenotypic (mixed transcriptomic subtypes) differences. From our limited explorations, we found that the signal of natural selection can be very weak, when we sample within the subpopulations. Patient not in the SCD distribution might represent cases of ongoing selection where natural selection has not yet yield large phenotypic differences. 4) The number of sectors sampled for each patient varies a lot, this can significantly influence the power of detecting natural selection.

Given the above points, we think the reviewer pointed to us an important gap in the field how to combine multi-omic (genotypic/phenotypic) evidence to jointly detect non-neutral evolution. We think this is a challenge facing the whole community. The study presented in our work provided the first glimpse into this interesting landscape and will require more efforts from the field and ourselves to construct a more systematic framework dissecting the genotypic/phenotypic heterogeneity and detect natural selection.

Revision: In light of the reviewers' comments, we have added above discussions in the "Discussions". Moreover, we have also extended the discussion in Supplementary Note 4.

We want to take this opportunity to thank the reviewer for the constructive comments. We have tried our best to address all the questions. These revisions have significantly improved the presentation of our work. We hope you find the revisions to your satisfaction.

References

1 Sun, R. *et al.* Between-region genetic divergence reflects the mode and tempo

of tumor evolution. *Nat Genet* **49**, 1015-1024 (2017).
<https://doi.org:10.1038/ng.3891>

REVIEWER COMMENTS

Reviewer #2 (Remarks to the Author):

The authors have addressed the two points I raised.

Reviewer #5 (Remarks to the Author):

Liu et al. propose an interesting experimental design to study the intratumor heterogeneity and evolution of hepatocellular carcinoma (HCC) in space, which they refer to as sITH.

So far, previous similar studies across cancers have been limited in the number of macroscopic regions and in how representative these regions were of the whole tumor mass.

To answer this, the authors profiled 222 macroscopic tumor "sectors" across 13 HBV+ HCC with RNA-seq and whole-exome sequencing (WES).

First, they found that the genomics landscape of their cohort recapitulates what was already shown in previous studies. Then they found that there was a spatial delineation of most subclones by sectors across HCC, which they refer to as "isolation-by-distance" (IBD), a term used in evolutionary genetics to qualify the presence of a physical-to-genetic distance relationship. To quantify IBD, they borrowed and adapted a score from population genetics, the fixation index (F_{ST}). IBD was also seen at the transcriptomic level, which they then annotate with subphenotypes extracted from the TCGA and driver events.

Overall, Liu et al. propose an interesting and unique dataset for the study of spatial ITH, which I think is worth publishing as a resource.

However, I have some reservations, akin to the ones from reviewer 3, on the statistics used and the conclusions made from the data by the authors.

I list them and elaborate below.

* Fig 2a. and Lines 187-188 "For example, two parallel linear relationships exist in patient SH05 and many nearby samples have surprisingly high genetic divergence (top left corner of Fig. 2a)."

I think this scatter plot and pointing that "nearby samples have surprisingly high genetic divergence" is not the best way to look at IBD. Even with perfect spatial segregation of subclones, which is the case here on panel e, one would expect close physical but far genetic distance at the border of different subclones. This is expected from a physical model of tumor growth. A better metric would quantify admixture with a local neighbourhood and normalise this score by the score of a perfect segregation vs. completely mixed model with the same phylogeny. In this case, SH05 would have a score equal to a perfect segregation.

* θ_R does not seem like it necessarily properly captures comparable local ITH, as it is based on non-shared mutations, which is subject to a strong technical component, i.e. the sequencing depth. This ratio should be corrected for sequencing depth.

* Lines 208-209

"Strikingly, the genetic differentiation between blocks is often much higher than genetic differentiation within blocks (Fig. 2d). Thus, the heterogeneity within HCC is spatially punctuated with regional homogeneity within blocks"

This jump in reasoning is really not clear to me. I think the term "punctuated" is being misused or should be redefined. What is punctuated heterogeneity and how does it differ from punctuated evolution?

How do the authors link between/within genetic differences to punctuatedness? There are other models that are gradual (non punctuated) that could explain low-within/high-between genetic divergence even at smaller histological scales, as is seen in many normal healthy tissues (where small clonal expansions are linked to non punctuated ageing/clock-related evolution).

* Lines 328-329

Using the classical imbalance index in phylogenetics known as the Sackin index S , we found significant evidence for clonal asymmetry for SH05 (P -value=0.0, Fig. 5b), suggesting Darwinian evolution driving the rapid expansion of the derived clone.

This is going a bit fast to conclusions again. Darwinian selection vs. neutrality are not the only factors influencing the balance of the tree (measured by the Sackin index), as it can simply be due to geographical constraints (see e.g. discussion here <https://doi.org/10.1111/ecog.04937>).

On specific aspects:

* Readability

To me the manuscript is easy to follow in its structure but is hard to read in the details. Essentially, English is approximate at times, especially in the discussion; and there are non sequiturs as already pointed out.

* Regarding the FST statistic,

I think the use of a genetic distance is well justified but the FST formulation would need to be adapted to the context of the study. In its current form, I do not think it properly measures what the authors claim (but could be a maladapted proxy for it where confounders are controlled for). Indeed, it should be made robust to confounding factors, especially purity and CNA.

Hudson's FST is meant to capture genetic distances between populations from SNP allele frequencies where d is the number of individuals sampled in each population. The way the authors defined it has a few issues: the translation from individuals (diploid samples => two phases measured out of two phases) to reads (single molecule in potentially non-diploid context => 1 phase out of X phases) is not appropriate; the raw VAF will be confounded by purity and CNA; the allele frequencies of SNVs do not behave as SNPs, notably because of CNAs and purity. Replacing the VAF by CCF values instead would make this formulation a better proxy of the genetic distance. However, I think the FST might not be an adapted formulation. The sum of branch lengths between tips of the sector tree would be a more adapted and tractable genetic distance in this case; and perhaps one to compare the FST to.

Lines 547.548

"The phylogenetic distance was extracted from the phylogenetic relationship. The clonal distance was calculated as the Euclidian distance between samples based on their clonal composition."

Please show the formal equation as this has multiple interpretations. Why Euclidean distance? To me, from the tree, the genetic distance would simply be the sum of the branch lengths (unique mutations to each most recent common ancestor of the two sectors).

* Use of phylogenetic distances vs. FST

As mentioned, the FST could use the CCFs instead but seems maladapted to tumor reads. Instead a phylogenetic distance extracted from the sector tree seems more appropriate (e.g. sum of branch lengths between sectors).

* Discrepancies between statistics FST vs. tree-based distance

I think the FST with CCF instead of VAF could be a relatively good proxy for the sum of branch lengths. The authors could show a scatter plot of these values across pairs of sectors within patients. However, again, the FST formulation would still be maladapted to this context.

* Need for simulations

I think the authors could not only use simulations to show that their FST metric accurately captures genetic distances even in the presence of confounders (purity, CNA, subclonal structure) but also: first to make their spatial growth models (and interpretations) explicit, and second, to validate the many claims/interpretations that are made throughout the paper.

Instead of resorting to simulations, the other option would be to tone down some of these interpretations/claims (i.e. on presence of Darwinian evolution and punctuatedness) and replace FST by tree-based distances (or at least show that FST recapitulates tree-based genetic distances, e.g. scatterplot as suggested).

Reply to the reviewers:

Reviewer #5 (Remarks to the Author):

Liu et al. propose an interesting experimental design to study the intratumor heterogeneity and evolution of hepatocellular carcinoma (HCC) in space, which they refer to as sITH. So far, previous similar studies across cancers have been limited in the number of macroscopic regions and in how representative these regions were of the whole tumor mass. To answer this, the authors profiled 222 macroscopic tumor "sectors" across 13 HBV+ HCC with RNA-seq and whole-exome sequencing (WES). First, they found that the genomics landscape of their cohort recapitulates what was already shown in previous studies. Then they found that there was a spatial delineation of most subclones by sectors across HCC, which they refer to as "isolation-by-distance" (IBD), a term used in evolutionary genetics to qualify the presence of a physical-to-genetic distance relationship. To quantify IBD, they borrowed and adapted a score from population genetics, the fixation index (FST). IBD was also seen at the transcriptomic level, which they then annotate with subphenotypes extracted from the TCGA and driver events. Overall, Liu et al. propose an interesting and unique dataset for the study of spatial ITH, which I think is worth publishing as a resource. However, I have some reservations, akin to the ones from reviewer 3, on the statistics used and the conclusions made from the data by the authors. I list them and elaborate below.

Reply: We are grateful to the reviewer for the encouraging feedback and acknowledgment of the significance of our research. Below, we have comprehensively addressed the comments and provided detailed explanations and clarifications.

Q1. Fig 2a. and Lines 187-188 "For example, two parallel linear relationships exist in patient SH05 and many nearby samples have surprisingly high genetic divergence (top left corner of Fig. 2a)."I think this scatter plot and pointing that "nearby samples have surprisingly high genetic divergence" is not the best way to look at IBD. Even with perfect spatial segregation of subclones, which is the case here on panel e, one would expect close physical but far genetic distance at the border of different subclones. This is expected from a physical model of tumor growth. A better metric would quantify admixture with a local neighbourhood and normalise this score by the score of a perfect segregation vs. completely mixed model with the same phylogeny. In this case, SH05 would have a score equal to a perfect segregation.

Reply: We want to thank the reviewer for the interesting suggestion. In this work, we aimed to dissect the genotypic and phenotypic landscape of HCC across space. With dense spatial sampling, we discovered block-shaped spatial heterogeneity within HCCs. For example, in patients such as SH05, we noticed that there are two parallel linear relationships between genetic divergence and physical distance between sectors (Figure 2a, maintext). We observed that within each spatial block there exist high spatial correlation resulting in the IBD pattern. On the other hand, the spatial correlation vanishes near the border of two spatial blocks and we observed "nearby samples have surprisingly high genetic divergence". We agree with the reviewer that the physical model of tumor growth can yield the pattern we discovered. The reviewer suggested us to develop a statistic to

quantify admixture within a local neighborhood by quantifying the amount of spatial correlation (i.e. perfect segregation with low spatial autocorrelation vs complete mixed model with high spatial autocorrelation).

Following the reviewer's suggestion, we employed a statistic from spatial statistics known as local Geary's C, which measures the local spatial autocorrelation across the two-dimensional space. It is defined as the ratio of local spatially-weighted squared distance to the total mean squared distance (see below for the exact formulation), which serves as a local indicator of spatial autocorrelation. In order to test the significance of the observed Geary's C values, we randomly shuffled the sectors across spatial locations of sectors (corresponding to the case of complete mixed model recommended by the reviewer) and re-calculated the local Geary's C value. By comparing the observed and permuted values (n=1000), we calculated the empirical p-values of the observed Geary's C. We found that sectors at the boarder of the spatial blocks tend to have very weak spatial autocorrelation (i.e. large p-value), whereas sectors within spatial block often show significant p-values with strong spatial autocorrelation (Figure 1, in this reply).

Figure 1: the empirical p-values of the local Geary's C calculated for SH05

Definition of Geary's C.

The local Geary's C index for sector i is defined as:

$$C_i = \frac{2n^2}{\sum_i \sum_j D_{ij}^2} \sum_j w_{ij} D_{ij}^2$$

where j represents sectors other than i , and n represents the total number of sectors in the patient. w_{ij} denotes the spatial weight between sector i and j . We employed the canonical form of w_{ij} by taking the inverse of the physical distance between sector i and

j. D_{ij} represents the genetic distance between sector i and j which we used the Hamming distance between mutations found in sector i and j .

To test the significance of the observed value, we calculated the empirical p-value of the observed local Geary's C. We randomly permuted the sectors spatially and recalculated the Geary's C and computed the proportion of replicates where the permuted c_i s are less than or equal to the observed C_i . In case of strong positive local spatial autocorrelation, the local Geary's C tends to have higher value than expected. On the contrary, regions with weak or negative spatial autocorrelation will result in small C_i .

Revision: we have now added the new results to the maintext and M&M (under section "Spatial correlation and the Geary's C statistic", on page 18) as well as a new Supplementary Figure 8.

Q2. θ_R does not seem like it necessarily properly captures comparable local ITH, as it is based on non-shared mutations, which is subject to a strong technical component, i.e. the sequencing depth. This ratio should be corrected for sequencing depth.

Reply: Thank you for the very careful thought. The reviewer is quite right that sequencing depth can influence the variant calling and subsequently might influence θ_R estimation. In the sequencing of the tumor sectors, we aimed to sequence all the sectors at a similar depth (i.e. mean depth of 100x), but in reality, there exist slight variation in the actual sequencing coverage.

Following the reviewer's comment, we down-sampled the sequencing depth of all the sectors to the minimum depth across all the sectors for a given patient. When we compared the number of mutations detected in the original sequencing depth (i.e. no downsampling) vs the number of mutations detected in the down-sampled scenario. We found highly similar number of mutations (Figure 2, this reply).

Figure 2: Number of detected mutations under the original coverage (x-axis) and number of mutations detected with the downsampled sequencing depth (all the sectors in a given patient were downsampled to the minimum coverage across all the sectors) (y-axis).

Thus, in the range of sequencing coverage we had, the number of mutations is not strongly affected by the small fluctuation in the sequencing coverage. When we recalculated the θ_R value with the downsampled mutation list, we observed highly similar results (Figure 3, this reply).

Figure 3: θ_R before and after downsampling. a) estimated θ_R before downsampling (x-axis) and after downsampling (y-axis), b) the estimated θ_R across space before downsampling, c) the estimated θ_R across space after downsampling.

Revision: we have added the results to the maintext (page 8), updated M&M (page 20) as well as a new Supplementary Figure 10.

Q3. Lines 208-209 "Strikingly, the genetic differentiation between blocks is often much higher than genetic differentiation within blocks (Fig. 2d). Thus, the heterogeneity within HCC is spatially punctuated with regional homogeneity within blocks" This jump in reasoning is really not clear to me. I think the term "punctuated" is being misused or should be redefined. What is punctuated heterogeneity and how does it differ from punctuated evolution? How do the authors link between/within genetic differences to punctuatedness? There are other models that are gradual (non punctuated) that could explain low-within/high-between genetic divergence even at smaller histological scales, as is seen in many normal healthy tissues (where small clonal expansions are linked to non punctuated ageing/clock-related evolution).

Reply: We want to thank the reviewer for the insightful thought. In the previous version of the work, we intended to use "punctuated heterogeneity" to describe the discontinuity in the spatial heterogeneity where tumors were spatially partitioned into spatial blocks with large genotypic/phenotypic differences between blocks, but small differences within blocks (i.e. block-shaped spatial heterogeneity).

Following the reviewer's comments, we did a bit literature search and "punctuatedness" (e.g. punctuated equilibrium/punctuated evolution) was often employed to describe "burst of activities" as contrast to stasis. Even though we were attempted to use large differences between blocks (e.g. between-cluster variance > 10 within-cluster variance) to "define" punctuatedness, as pointed out by the reviewer, pinpointing a definitive cutoff to define "punctuatedness" is not as straightforward and can be subjective. In light of the reviewer's

constructive suggestions presented in the later questions (See later question in MQ6), we have followed your advice to remove the “punctuatedness” from the latest version of the manuscript.

Revision: We have removed the term punctuatedness from the latest version of the manuscript including 1) removing the argument around punctuatedness in section 3 and 4 and have updated the subtitles to “Block-shaped genotypic heterogeneity in HCC” and “Block-shaped phenotypic heterogeneity mirrors the genotypic heterogeneity” (page 7-8). 2) we have also removed the punctuatedness from the title of our work and have updated our title to “Tumor phylogeography revealed block-shaped spatial heterogeneity and the mode of evolution in Hepatocellular Carcinoma”. We want to thank the reviewer for the constructive suggestion.

Q4. Lines 328-329 Using the classical imbalance index in phylogenetics known as the Sackin index S , we found significant evidence for clonal asymmetry for SH05 (P-value=0.0, Fig. 5b), suggesting Darwinian evolution driving the rapid expansion of the derived clone. This is going a bit fast to conclusions again. Darwinian selection vs. neutrality are not the only factors influencing the balance of the tree (measured by the Sackin index), as it can simply be due to geographical constrains (see e.g. discussion here <https://doi.org/10.1111/ecog.04937>).

Reply: Thank you for pointing out the reference to us. We took a careful read of the reference and other related work in the literature. We agree with the reviewer that Darwinian/Neutral evolution is not the sole factor contributing to the observed spatial heterogeneity. As pointed out by the reviewer, geographic isolation/physical constraint can also be an important factor contributing to clonal asymmetry.

Revision: Following the reviewer’s constructive suggestion (See later question in MQ6), we have 1) attenuated the conclusion regarding Darwinian selection in the cohort. We have concluded that many evolutionary forces including geographic isolation/physical constraint can also contribute to the block-shaped heterogeneity as well as clonal asymmetry (page 14). 2) we have changed our title to “Tumor phylogeography revealed block-shaped spatial heterogeneity and the mode of evolution in Hepatocellular Carcinoma”. 3) we have also discussed possible factors contributing to the observed spatial heterogeneity in the revised discussion (page 14).

On specific aspects:

MQ1. Readability

To me the manuscript is easy to follow in its structure but is hard to read in the details. Essentially, English is approximate at times, especially in the discussion; and there are non sequiturs as already pointed out.

Reply: Thank you for the comments. We have followed the reviewer’s advice to: 1) provide more details in M&M about the methods (page 17-22); 2) we have rewritten the discussion, especially removing the speculative points (page 14).

MQ2.Regarding the FST statistic,

I think the use of a genetic distance is well justified but the FST formulation would need to be adapted to the context of the study. In its current form, I do not think it properly measures what the authors claim (but could be a maladapted proxy for it where confounders are controlled for). Indeed, it should be made robust to confounding factors, especially purity and CNA. Hudson's FST is meant to capture genetic distances between populations from SNP allele frequencies where d is the number of individuals sampled in each population. The way the authors defined it has a few issues: the translation from individuals (diploid samples => two phases measured out of two phases) to reads (single molecule in potentially non-diploid context => 1 phase out of X phases) is not appropriate; the raw VAF will be confounded by purity and CNA; the allele frequencies of SNVs do not behave as SNPs, notably because of CNAs and purity. Replacing the VAF by CCF values instead would make this formulation a better proxy of the genetic distance. However, I think the FST might not be an adapted formulation. The sum of branch lengths between tips of the sector tree would be a more adapted and tractable genetic distance in this case; and perhaps one to compare the FST to.

Reply and revision: Thank you for the thoughtful discussion on the proper formulation of FST. Even though we have adopted an approach implemented in a recent work which have taken into consideration of confounding issue of tumor purity and ploidy (e.g. based on cancer cell fraction), we very much agree with several of the concerns the reviewer raised, especially regarding lacking of proper consideration of the fact that the pooled reads from the tumor is not equivalent to the individual alleles called from a set of diploid individuals. Developing a proper statistical approach taking into consideration of these factors will require substantial efforts which is beyond the short revision time. In light of the reviewer's constructive suggestion (MQ6), we have taken the advice to remove FST from the latest version of our presentation.

MQ3. Lines 547.548

"The phylogenetic distance was extracted from the phylogenetic relationship. The clonal distance was calculated as the Euclidian distance between samples based on their clonal composition." Please show the formal equation as this has multiple interpretations. Why Euclidean distance? To me, from the tree, the genetic distance would simply be the sum of the branch lengths (unique mutations to each most recent common ancestor of the two sectors).

Reply: Thank you for the nice suggestions. In exploring the isolation-by-distance relationship, we have employed multiple statistics. For example, we have previously used a) FST (in the new version, we have replaced it with genetic distance), b) the phylogenetic distance, c) clonal distance in measuring genetic differentiation between sectors. Across multiple metrics, we found that there is a general isolation by distance relationship across multiple metrics (Supplementary figure 4).

The clonal distance is measured as the differences in the clonal composition. To be more precise, if we assume there are n clones presented in two sectors with proportions (p_1 ,

$p_2 \dots p_n$) and $(q_1, q_2, q_3 \dots q_n)$, we calculated the distance in the clonal composition between two samples as $\sqrt{\sum_i^n (p_i - q_i)^2}$. Take together, by using multiple distance metrics, we found that the isolation-by-distance relationship is consistent under a wide variety of distance metrics.

Revision: we have added the details of the formulation to the latest version of the M&M (page 18).

MQ4. Use of phylogenetic distances vs. FST

As mentioned, the FST could use the CCFs instead but seems maladapted to tumor reads. Instead a phylogenetic distance extracted from the sector tree seems more appropriate (e.g. sum of branch lengths between sectors).

Reply and revision: We agree with the reviewer and we have removed FST from our latest presentation and (see MQ2) and have substitute FST with genetic distance recommended by the reviewer (also see MQ2 and MQ6).

MQ5. Discrepancies between statistics FST vs. tree-based distance

I think the FST with CCF instead of VAF could be a relatively good proxy for the sum of branch lengths. The authors could show a scatter plot of these values across pairs of sectors within patients. However, again, the FST formulation would still be maladapted to this context.

Reply and revision: thank you for nice comments. As we have shown in our previous version, we have employed multiple metrics including FST and genetic distance (i.e. tree-based distance), we found that multiple genetic distance metrics are highly correlated and all showed a strong positive correlation with physical distance. In the updated version, we have removed FST from the latest version, we have updated the presentation with the new distance metric (Supplementary 4)

MQ6. Need for simulations

I think the authors could not only use simulations to show that their FST metric accurately captures genetic distances even in the presence of confounders (purity, CNA, subclonal structure) but also: first to make their spatial growth models (and interpretations) explicit, and second, to validate the many claims/interpretations that are made throughout the paper.

Instead of resorting to simulations, the other option would be to tone down some of these interpretations/claims (i.e. on presence of Darwinian evolution and punctuatedness) and replace FST by tree-based distances (or at least show that FST recapitulates tree-based genetic distances, e.g. scatterplot as suggested).

Reply and revision: We really appreciate the kind suggestion from the reviewer and we agree that it can be important to perform simulations to test out the performance of the FST metric under the influence of confounding factors such as tumor purity, CNA and subclonal structure. However, considering it is substantial amount of effort to write a good computer program to simulate realistic tumor growth and test out FST under a variety of scenarios in the short revision time. Instead, we have followed the reviewer's advice to a) replace FST with genetic distances (MQ3/MQ4/MQ5), b) remove the punctuatedness (Q3), c) tone

down the claim on presence of Darwinian selection (Q4). We really appreciate the constructive suggestions from the reviewer. We think these revisions have substantially increased the quality of our presentation and we will further pursue some of the questions in a future study.

REVIEWERS' COMMENTS

Reviewer #5 (Remarks to the Author):

I thank the authors for the thorough replies to my comments and wish to congratulate them on the work. I have no further comments.

Reviewer #5 (Remarks on code availability):

Regarding the github, it seems most of the code is there for the analyses and even for figure generation. However, the README is empty, which makes reproducibility impossible.

The authors should provide at least:

- a description of the directory structure,
- a description of each script,
- instructions for installation including dependencies
- example code to run each script
- description of inputs and outputs and ideally example input data for each script.

Reviewer #5 (Remarks to the Author):

I thank the authors for the thorough replies to my comments and wish to congratulate them on the work. I have no further comments.

Reviewer #5 (Remarks on code availability):

Regarding the github, it seems most of the code is there for the analyses and even for figure generation. However, the README is empty, which makes reproducibility impossible.

The authors should provide at least:

- a description of the directory structure,
- a description of each script,
- instructions for installation including dependencies
- example code to run each script
- description of inputs and outputs and ideally example input data for each script.

Reply: We have corrected our code availability as suggested.